# Identification of type VI secretion system effector-immunity pairs using structural bioinformatics

Alexander M Geller [ID] [1], Maor Shalom[1,2], David Zlotkin[1,2], Noam Blum[1] & Asaf Levy [ID] [1✉]

## Abstract

The type VI secretion system (T6SS) is an important mediator of microbe–microbe and microbe–host interactions. Gram-negative bacteria use the T6SS to inject T6SS effectors (T6Es), which are usually proteins with toxic activity, into neighboring cells. Antibacterial effectors have cognate immunity proteins that neutralize self-intoxication. Here, we applied novel structural bioinformatic tools to perform systematic discovery and functional annotation of T6Es and their cognate immunity proteins from a dataset of 17,920 T6SS-encoding bacterial genomes. Using structural clustering, we identified 517 putative T6E families, outperforming sequence-based clustering. We developed a logistic regression model to reliably quantify protein–protein interaction of new T6E-immunity pairs, yielding candidate immunity proteins for 231 out of the 517 T6E families. We used sensitive structure-based annotation which yielded functional annotations for 51% of the T6E families, again outperforming sequence-based annotation. Next, we validated four novel T6E-immunity pairs using basic experiments in *E. coli*. In particular, we showed that the Pfam domain DUF3289 is a homolog of Colicin M and that DUF943 acts as its cognate immunity protein. Furthermore, we discovered a novel T6E that is a structural homolog of SleB, a lytic transglycosylase, and identified a specific glutamate that acts as its putative catalytic residue. Overall, this study applies novel structural bioinformatic tools to T6E-immunity pair discovery, and provides an extensive database of annotated T6E-immunity pairs.

**Keywords** Alphafold-multimer; Effector-immunity Pairs; Foldseek; Structural Bioinformatics; Type VI Secretion System (T6SS)
**Subject Categories** Computational Biology; Microbiology, Virology & Host Pathogen Interaction; Structural Biology

## Introduction

Microbes utilize various antagonistic mechanisms to infect hosts and to kill competing microbes. The type VI secretion system (T6SS) is a membrane-bound, contact-dependent secretion system that injects toxic T6SS effector proteins (T6Es) into neighboring

bacteria and into eukaryotic cells (Cianfanelli et al, 2016; Pukatzki et al, 2006; Trunk et al, 2018; Coulthurst, 2013; Le et al, 2021; Jurėnas and Journet, 2021). Structurally, the T6SS is a contractile injection system, which shoots a spear-like structure toward neighboring cells. The tip and shaft are secreted from the attacking cell (Vettiger and Basler, 2016). The tip is made up of a protein called PAAR (proline–alanine–alanine–arginine), and a trimer of VgrG (valine–glycine repeat protein G), while the shaft is made up of a column of hollow ring structures, formed from stacks of hexamers of the protein Hcp (Hemolysin-coregulated protein) (Wang et al, 2019; Shneider et al, 2013).

T6Es are oftentimes proteins that non-covalently interact with Hcp, VgrG, or PAAR, and are thereby delivered upon T6SS contraction and penetration into target cells. These T6Es are called "cargo" effectors (Alcoforado Diniz et al, 2015; Monjarás Feria and Valvano, 2020). There are also "specialized effectors", which contain an N-terminal Hcp, VgrG, or PAAR, and a C-terminal toxin domain (Alcoforado Diniz et al, 2015; Coulthurst, 2019; Ma et al, 2009; Hespanhol et al, 2022). An example of a specialized effector from *Vibrio cholerae* encodes for a protein with an N-terminal VgrG and a C-terminal actin crosslinking domain (Pukatzki et al, 2007). Other examples of specialized effectors include VgrG-3 of *V. cholerae*, which has a C-terminal peptidoglycan degrading activity (Brooks et al, 2013), VgrG2B of *Pseudomonas aeruginosa*, which has a C-terminal metallopeptidase activity (Wood et al, 2019), Hcp-ET1, which has a C-terminal nuclease domain (Ma et al, 2017), and Tse6 of *Pseudomonas aeruginosa* has an N-terminal PAAR with a C-terminal NAD(P)(+) degrading toxin (Whitney et al, 2015; Quentin et al, 2018).

Because T6Es many times target essential structures of prokaryotes, producing cells must prevent T6Es from causing self-intoxication or from toxicity via injection by sister cells. This is usually carried out using immunity proteins, which are frequently encoded downstream of T6Es, and which bind and neutralize the T6Es toxic activity (Allsopp and Bernal, 2023; Berni et al, 2019; Hood et al, 2010). However, resistance to T6E toxicity independent of immunity proteins has also been described (Le et al, 2020; Hersch et al, 2020a, 2020b). T6E-immunity pairs are actively researched because the T6SS is important to microbial ecology via their key role in niche colonization and pathogen–host interaction (Speare et al, 2018; Logan et al, 2018; Dörr and Blokesch, 2018; Mosquito et al, 2020). Furthermore, T6Es can be developed into potential new antimicrobials for both medical and agricultural applications (Jana et al, 2021; Borrero de Acuña and Bernal, 2021).

[1]Department of Plant Pathology and Microbiology, The Institute of Environmental Science, The Robert H. Smith Faculty of Agriculture, Food and Environment, The Hebrew University of Jerusalem, Rehovot, Israel. [2]These authors contributed equally: Maor Shalom, David Zlotkin. ✉E-mail: alevy@mail.huji.ac.il

Despite this importance, a survey screened Proteobacterial genomes that contain T6SS genes and found that only 42% had effectors of known activity, which suggests a large portion of effectors remain undiscovered or their mechanisms of action are yet unknown (LaCourse et al, 2018). Many T6E-immunity gene pairs remain hidden in microbial genomes, and new methods are required to accurately discover these genes and decipher their molecular functions.

Recently, there was a breakthrough in protein structure prediction by Alphafold2. The program takes amino acid sequences as input and outputs predicted protein structures along with quantitation of confidence in the prediction (Jumper and Hassabis, 2022; Jumper et al, 2021). Excitingly, modest adjustments to Alphafold2 resulted in a deep learning model called AlphaFold-Multimer that can not only predict accurate three-dimensional structures of monomers, but also of multimers (Jumper et al, 2021; Jumper and Hassabis, 2022; Tsaban et al, 2022; Evans et al, 2022; Yin et al, 2022; Bryant et al, 2022a, 2022b; Gao et al, 2022). Because T6E-immunity pairs tightly bind to one another, we surmised that Alphafold-Multimer could be applied to predict novel T6E-immunity pairs. The creation of millions of novel predicted protein structures in bioinformatic databases online prompted the development of Foldseek (Hutson, 2023). Foldseek is a protein structure-based search algorithm that can rapidly and efficiently search millions of predicted and experimentally determined protein structures (Hutson, 2023). By analogy, it is like BLAST (Camacho et al, 2009; Altschul et al, 1990), but for protein structures. Structural search is much more sensitive and could provide useful insights into the functions of putative T6Es as compared to using amino acid-based search alone. Alphafold-Multimer was recently used to explore toxin-immunity networks (Ernits et al, 2023), but to the best of our knowledge, these structural bioinformatic tools have not been systematically used to discover T6Es, their immunity proteins, and their functions on a large scale.

In this study, we performed a broad-scale bioinformatic analysis on 17,920 T6SS-encoding genomes, with a specific focus on specialized Hcp, VgrG, and PAAR effectors. We utilized novel structural bioinformatic tools for T6E clustering, prediction of T6E-immunity binding, and structure-based search. (Evans et al, 2022; van Kempen et al, 2024; Tsaban et al, 2022; Yin et al, 2022). As a resource for the community, we provide annotations for 265 out of the 517 specialized T6E domain families. Of these annotated clusters, Foldseek provided the annotation information for about two-thirds, while Pfam provided annotation for only one-third. Furthermore, our logistic regression classifier showed that 231 out of 517 T6E domain families have a predicted immunity protein based on modeled protein–protein interactions. To show the accuracy of our algorithm, we confirmed the activity of four putative T6E-immunity pairs in vitro by demonstrating that the T6E is toxic to *E. coli* cells and the cognate immunity proteins neutralize the T6E toxicity. We propose that DUF3289 is a homolog of colicin M, and that DUF943 acts as its cognate immunity protein. We show that a homolog of lytic transglycosylase SleB is a putative T6E and that its expression in Gram-negative bacteria leads to intriguing double-membrane cells. This proof-of-concept not only provides an organized database of hundreds of specialized T6E and immunity families, but also more broadly highlights the power of structural bioinformatic tools in T6E gene discovery and functional annotation.

# Results

## Structure-based T6E clustering is more sensitive than sequence-based clustering

Our initial T6E dataset was constructed by searching the Integrated Microbial Genomes & Microbiomes (IMG/M) database (Chen et al, 2019, 2017; Markowitz et al, 2012) for genes that match the modular structure of specialized T6Es. Specifically, we looked for genes with an N-terminal T6SS core domain (PAAR, VgrG, Hcp, and PAAR-like domain DUF4150), along with an extended C-terminal domain, which likely acts as a toxic effector domain (Figs. 1A and EV1). We specifically chose specialized effectors because they are highly likely to be bona fide T6Es, due to their physical linkage to N-terminal structural T6SS domains, and can therefore be used as an easily detectable genetic signature of T6Es. We identified genes that encode for proteins with N-terminal T6SS core domains from a database of 17,920 T6SS-encoding genomes. Then, we clustered them into full-length protein clusters using sequence-based clustering ("Methods"). To find specialized effectors, we took representatives from each full-length protein cluster and filtered for those with C-terminal extensions beyond the core structural Pfam domains ("Methods"), leaving 1192 putative specialized T6Es. We then removed the N-terminal cores, thereby extracting only the putative C-terminal putative effector domain. These domains were then clustered using sequence-based methods into 1065 unique C-terminal domain clusters (Fig. 1A).

In a typical bioinformatic analysis, the 1065 sequence-based clusters would be considered the terminal step in clustering of these C-terminal domains. Here, we extended this analysis by taking the sequences of the representatives of these 1065 domains and predicting their three-dimensional protein structures using Alphafold2 (Jumper et al, 2021; Jumper and Hassabis, 2022). Then, we used the novel structure-structure comparison algorithm Foldseek to cluster the structures, overcoming the sequence-based barrier, resulting in 517 structure-based clusters, likely representing the structure space of specialized T6Es in Proteobacteria in the IMG/M database (Fig. 1A,B). The structure-based clusters consisted of 157 non-singleton clusters, and 360 singleton clusters. This compression from 1065 sequences to 517 structures is in line with the fact that two divergent sequences with little to no sequence similarity can produce a similar structure in 3D space (Laurents et al, 1994). For example, 18 separate C-terminal sequence clusters were unified into one structural cluster, named Foldseek cluster 145 (Fig. 1B). Indeed, it is visually apparent that their predicted protein structures share a similar domain (Fig. 1C, purple highlight). Even with a more lenient lower bound for sequence-based clustering of 30% identity (in the "twilight zone" of sequence identity), a compression of 1065 sequences into only 845 C-terminal clusters was seen. This shows that structural clustering indeed surpasses the abilities of sequence-based clustering. Overall, we used structure-based clustering to classify specialized effector C-termini into 517 families, which in total represents 21,939 full-length proteins in 282 genera in the IMG/M database.

## Alphafold-Multimer-based regression model accurately predicts T6E-immunity pairs

Alphafold2 not only predicts protein structures of single amino acid sequences, but was also augmented to predict the folding of

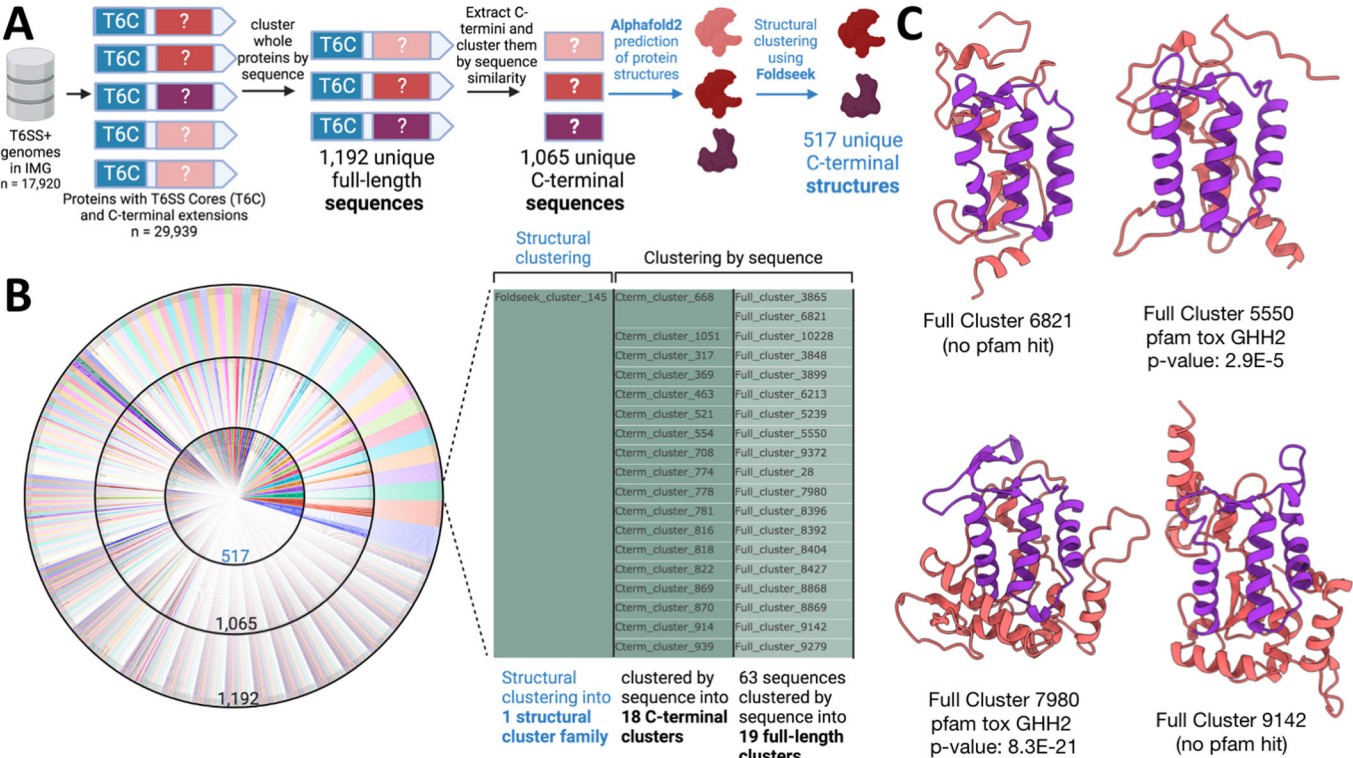

**Figure 1. Structural clustering reveals 517 C-terminal T6E domain families.**

(A) Pipeline for mining and clustering specialized T6E C-termini toxin domains. From the IMG/M database, 17,920 T6SS-encoding genomes were queried for genes encoding for proteins with N-terminal T6SS core domains (e.g., PAAR). Those with long C-termini were considered specialized effectors (*n* = 29,939). The proteins were then clustered using sequence-based methods into 1192 clusters. Then, the N-termini core domains were removed, and the C-termini alone were clustered using sequence-based techniques into 1065 clusters. Each sequence had its structure predicted by Alphafold2, and then clustered using the structure-based algorithm of Foldseek into 517 clusters. (B) A sunburst plot showing the full-length clusters (outermost ring), the C-terminal clusters (middle ring), and the structural clusters (inner ring). Foldseek cluster 145 is shown enlarged; (C) predicted protein structures of four select members of this cluster are shown on the right (two that have a Pfam annotation with an associated p value, and two that do not). Purple regions highlight a core conserved domain between the protein structures.

multiple amino acid chains simultaneously. This deep learning model, called Alphafold-Multimer, provides metrics that estimate the quality of folding of two proteins together in three-dimensional space, which can be used to determine if two proteins interact (Evans et al, 2022). We therefore hypothesized that we could use Alphafold-Multimer to quantify the interaction between putative T6Es and their putative cognate immunity proteins. A stable interaction raises our confidence that the predicted gene is indeed a toxic T6E that requires a cognate immunity protein. This approach has recently been used successfully in a type II toxin–antitoxin study (Ernits et al, 2023), but has not been systematically applied to T6SS.

To test whether Alphafold-Multimer could provide a reliable quantitative signal of binding between T6Es and their cognate immunity proteins, we used a dataset of known, experimentally validated T6E-immunity pairs from the SecReT6 database (Fig. EV2A) (Zhang et al, 2023). We took these 95 cognate T6E-immunity pairs and predicted their dimeric structures using Alphafold-Multimer. Along with the protein structures, a confidence score in the protein–protein interaction called interface pTM score (ipTM) is outputted. ipTM has been shown to be effective in quantifying the confidence of interactions between proteins (Yin et al, 2022; Teufel et al, 2023). To confirm this, we

compared ipTM to alternative metrics like pDockQ (Bryant et al, 2022a) and mean intermolecular PAE (which is based on another Alphafold-Multimer output) alone and in combination, and found that ipTM alone was the best metric for predicting protein–protein interactions (Fig. EV2B). Among the cognate T6E-immunity pairs, we indeed observed high ipTM scores. To contextualize these scores, we compared these scores to a control group of shuffled T6E-immunity pairs, i.e., randomized, noncognate pairs. We then trained a machine learning classifier (a logistic regression model) to see if the ipTM confidence scores from the cognate pairs (positive set; size = 95) could be unbiasedly differentiated from noncognate random pairs (negative set; size = 69). We performed training using a 70–30% training–testing split (Fig. EV2A). We found that the test set Accuracy was 90% with an AUC of 0.914 (Figs. 2A and EV2C). This suggests that this trained logistic regression model (and the underlying ipTM data) is a reliable tool that can classify whether a T6E interacts with a putative immunity protein.

To apply this model to our data, we first needed to search our database of specialized T6E C-terminal clusters to identify the genes encoding their putative immunity proteins. Specifically, we chose genes that were (a) small (<700 amino acids), and (b) directly downstream of the putative T6Es (<=100 nucleotides), and (c) encoded on the same DNA strand with the T6E gene. Namely, we

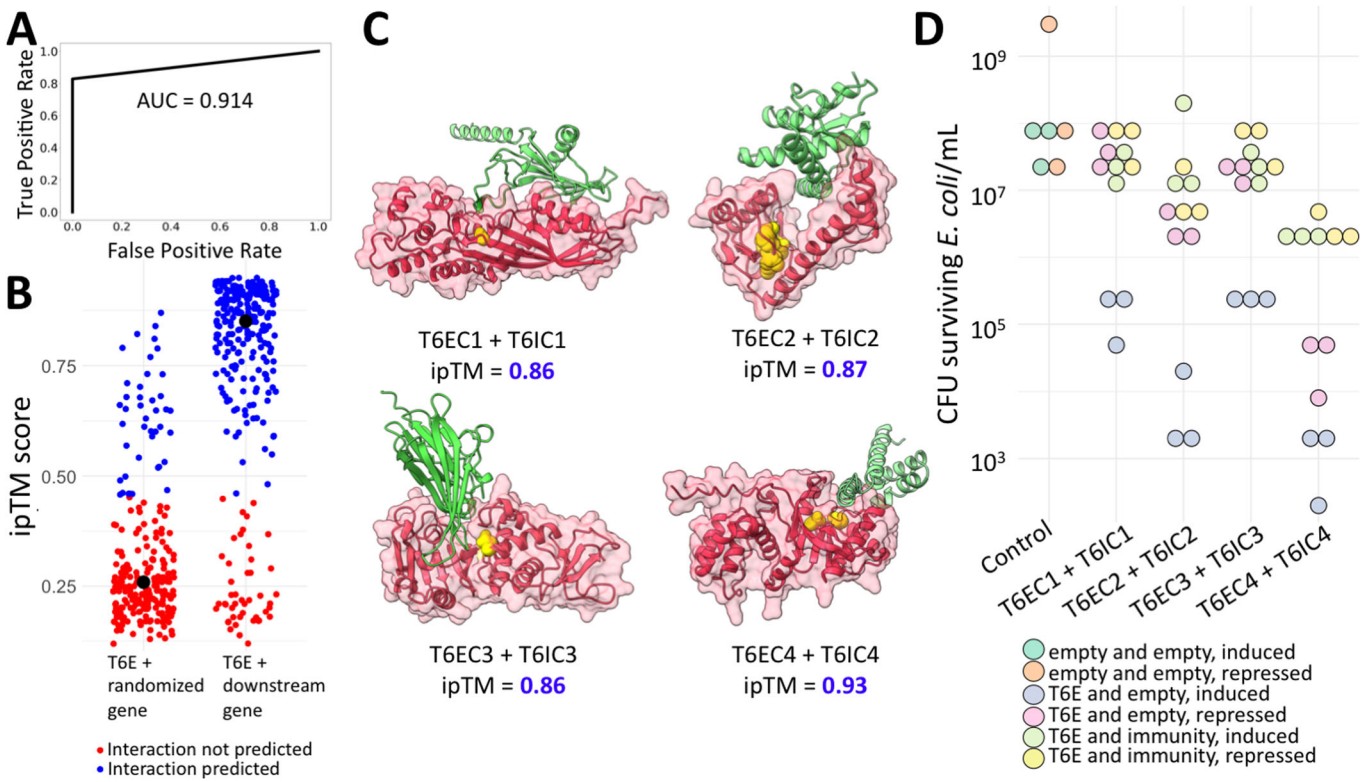

**Figure 2. T6E-immunity pairs are predicted by the ipTM score.**

(A) The Receiver Operating Characteristic plot for our trained logistic regression model, as evaluated on the test dataset. The model was trained on experimental T6E-immunity data, and is based on Alphafold-Multimer's ipTM score. The area under the curve measurement is 0.914. (B) The trained logistic regression model was used to investigate T6Es and adjacent, downstream genes encoded in the same direction as possible immunity proteins ("T6E + downstream gene"). The blue dots represent True labels by the model, i.e., predicted protein–protein interaction. The red dots represent False labels by the model, i.e., no predicted protein–protein interaction. As a control, T6Es and downstream genes were shuffled to make noncognate pairs ("T6E + randomized gene"). The black dot represents the median value. (C) The predicted protein structures of T6EC1-4 are shown in red, with yellow marking conserved residues, some of which are known catalytic residues. The green protein structure (and its position relative to the putative T6E) is the predicted immunity protein, as predicted by Alphafold-Multimer. (D) Quantification of a drop assay of *E. coli* BL21 heterologously expressing T6EC1-4 in pBAD24 (arabinose induction) and T6IC1-4 in pET29b (IPTG induction), or T6EC1-4 in pBAD24 and an empty pET29b. T6EC1-4 is either repressed (0.01 mM IPTG, 1% Glucose) or induced (0.01 mM IPTG, 0.2% Arabinose). The immunity genes are constitutively expressed in these experiments. T6EC1 and T6EC3 were expressed in the periplasm using an N-terminal twin-arginine translocation (TAT) sequence, and T6EC2 and T6EC4 were expressed in the cytoplasm. T6IC1 also had an N-terminal TAT sequence added, while T6IC3 already had a naturally occurring periplasmic trafficking signal. Each dot represents a biological replicate. A Dunn test was performed between the "T6E and empty, induced" versus the "T6E and immunity, induced" groups, with the null hypothesis meaning the presence of the immunity protein has no effect on toxicity. The results of the Dunn test reject the null hypothesis ($P = 0.023$, 0.023, 0.018, 0.023 for T6EC1-4, respectively), supporting the notion that the immunity protein neutralizes T6E toxicity. Source data are available online for this figure.

looked for signs that the T6E and its downstream gene are co-expressed. These genomic signatures make these prime candidates for encoding for immunity proteins (Allsopp and Bernal, 2023). We took 298 putative T6E-immunity pairs and input them into the trained logistic regression model, and found that 83% were classified as interacting (Fig. 2B, "T6E + downstream gene"). Importantly, 237 noncognate, shuffled putative T6E-immunity pairs only saw 16.9% classified as interacting) (Fig. 2B, "T6E + randomized gene"). This suggests we can use ipTM scores to define these adjacent genes that have genomic signatures of T6E-immunity pairs as immunity proteins. Based on this definition, out of the 517 putative T6E clusters, 231 had at least one predicted immunity protein (Supplementary Data 1).

To experimentally validate the hypothesis that these are indeed T6E-immunity pairs, we tested whether expression of four putative T6Es were toxic to *Escherichia coli*, and whether their cognate immunity proteins could rescue *E. coli* from toxicity. We named

these four T6E Candidates 1–4 (T6EC1–4). All four putative T6E-immunity pairs had high ipTM scores, and were predicted to interact in a manner where the immunity protein blocks access to catalytic/conserved residues in the effector protein (Fig. 2C). T6EC1 is encoded in *Yersinia pseudotuberculosis* IP32881, a species that causes foodborne illness in humans (Brady et al, 2022). T6EC1 is a specialized PAAR encoded in a T6SS operon, indicating its likely role as a T6E. Because the T6IC1 putative immunity gene was predicted to encode a transmembrane anchoring domain with the bulk of the protein facing the periplasm, we hypothesized that the activity of T6EC1 is in the periplasm (Fig. EV3A). T6EC2 is found in *Herbaspirillum seropedicae* Os45, a rice pathogen (Zhu et al, 2012). T6EC2 has an N-terminal DUF4150 (PAAR-like) domain, and an unannotated C-terminal domain (Fig. EV3B). T6EC3 is encoded in *Acinetobacter rudis* CIP 110305, a microbe that was isolated from raw milk (Vaz-Moreira et al, 2011). It has an N-terminal PAAR domain, and a C-terminal domain with no

annotation (Fig. EV3C). Using signalP 6.0, we identified that the immunity protein T6IC3 has a predicted periplasmic localization signal, suggesting that the toxin has activity in the periplasm. T6EC4 is encoded in *Citrobacter koseri* FDAARGOS_164, a medically important bacteria that was isolated from a urine catheter (Sichtig et al, 2019). T6EC4 has an N-terminal Hcp domain and a C-terminal unannotated domain (Fig. EV3D). We heterologously expressed T6ECs on pBAD24 expression vectors, paired either with a pET29 vector expressing the putative cognate immunity protein or with an empty pET29 vector as a control. Because their immunity proteins were predicted to be active in the periplasm, expression of T6EC1 and T6EC3 was carried out in the periplasm using a twin-arginine translocation (TAT) sequence. We saw that expression of T6Es paired with an empty vector was toxic to *E. coli*, while expression of T6Es paired with a plasmid expressing immunity proteins prevented the toxic phenotype (Fig. 2D). These initial results are in line with the model of T6E-immunity binding, providing evidence that these are putative T6E-immunity pairs that bind one another. Further tests that are outside the scope of this study need to be performed to show that these are indeed T6E-immunity pairs, e.g., T6SS-dependent effector secretion and inter-microbial competition assays using different mutants. Overall, we propose that Alphafold-Multimer's ipTM score provides a reliable quantitative measure of interaction

between predicted T6E-immunity pairs, which can be used to complement genomic signatures for higher confidence data mining and can facilitate easier identification of T6E-immunity pairs in the laboratory. The trained logistic regression model is provided in "Methods".

## Foldseek structural search expands Pfam-based annotations of T6E function

To computationally determine possible functions of protein sequence databases, Pfam protein domain annotation is commonly used. We searched the sequences of the T6E C-terminal clusters versus the Pfam database and found 77 unique Pfam domains, the most frequent being Tox-GHH2, followed by Pyocin_S, and GH_E, which are all DNAse domains (Fig. 3B). However, this method only covers a small portion of the data (Fig. 3A) while a majority of the data lacks any Pfam annotation. To expand the annotations of the T6Es, we explored the ability of Foldseek to search large protein structure databases to annotate the 517 T6E structural clusters. Foldseek search (Barrio-Hernandez et al, 2023) was used to search the Alphafold Structure Database (Varadi et al, 2022), Uniprot (UniProt Consortium, 2019), and PDB100 (Burley et al, 2023), and significant alignments to annotated proteins were used to carefully manually classify each structural cluster. Overall, this resulted in

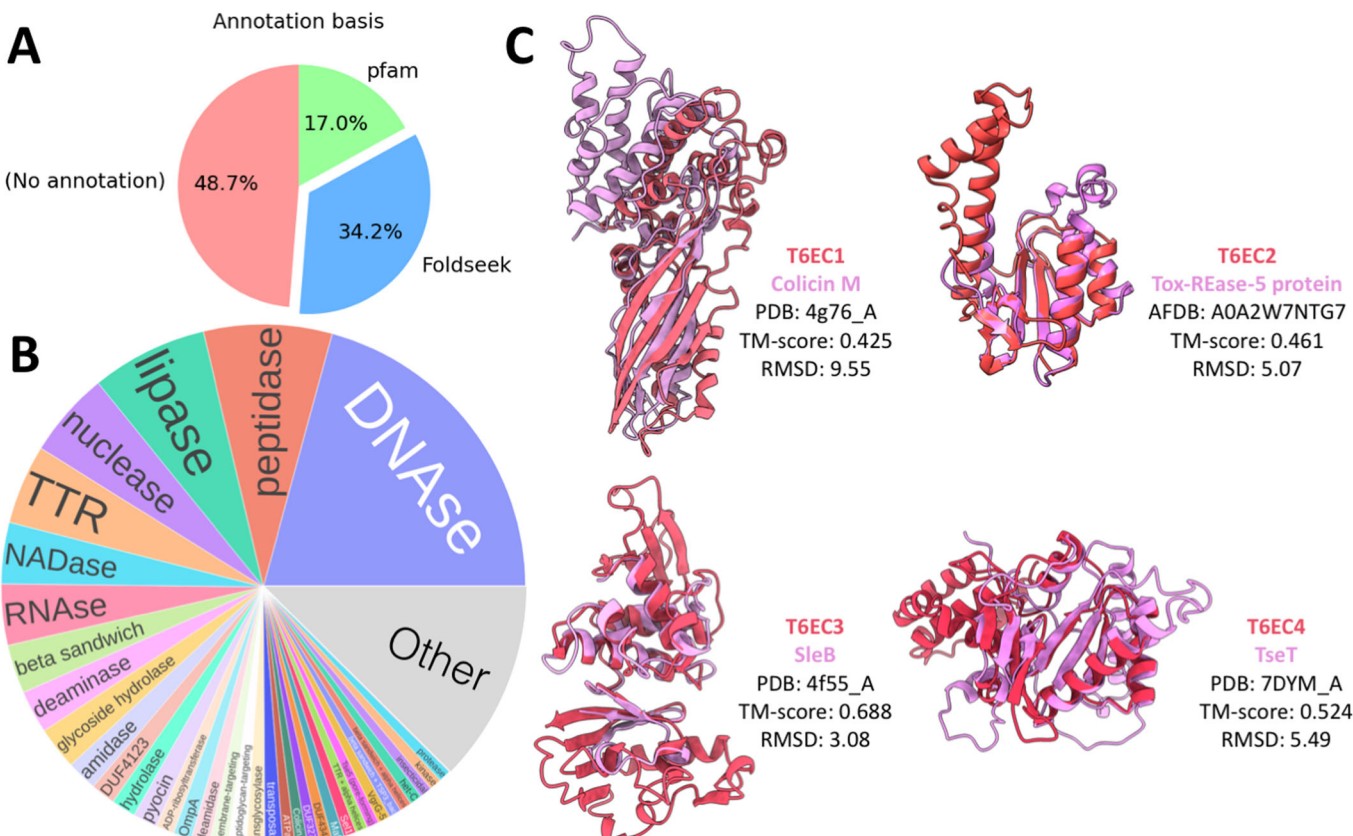

**Figure 3. Structural search outperforms Pfam for annotation of putative T6Es.**

(A) Pie chart showing the method that was ultimately used for annotation of the 517 clusters. (B) Annotations of the 265 annotated clusters. (C) T6ECs (red structures) and their homologs (purple structures).

265/517 (51%) clusters with annotation. While Pfam annotations accounted for ~17% (88/517) of annotations, Foldseek-based search results annotated ~34% (177/517) of structural clusters, constituting a significant advantage (Fig. 3A; Supplementary Data 1). The annotations revealed that most of the specialized T6Es are predicted to function as DNAses (Fig. 3B). Other domains included other enzymatic activities seen in other T6E effectors like lipase, NADase, peptidase, glycoside hydrolase, deaminase, deamidase, and ADP-ribosyltransferase (Allsopp and Bernal, 2023; Sibinelli-Sousa et al, 2020; Jana and Salomon, 2019; Hagan et al, 2023; Hernandez et al, 2020; de Moraes et al, 2021; Aubert et al, 2016; Whitney et al, 2013). Some C-terminal domains were not necessarily toxins, but had a Transthyretin domain (TTR), which is known for protein–protein interaction, as is seen in VgrG binding to Tle1 (Flaugnatti et al, 2020). The TTR domains may therefore bind cargo effectors. Interestingly, DUF4123, a known T6SS chaperone domain (Liang et al, 2015; Unterweger et al, 2015), was also found as a C-terminal domain of specialized T6SS effectors. However, pore-forming activity, which was described before for several T6Es, is relatively rare in our annotated dataset (Mariano et al, 2019; Ali et al, 2023; González-Magaña et al, 2022).

We asked whether the Foldseek-based annotations of T6EC1-4 from the previous section indeed provide reliable hypotheses for their mechanisms of action. T6EC1, a member of Foldseek cluster 294, had no functional annotation based on its sequence except for Pfam annotations as Domains of Unknown Function DUF3289. The same is true for its cognate immunity gene, T6IC1, which only has a DUF943 annotation (Fig. EV3A). In other words, these domains have been noted for their frequent appearance in genomes, but have no annotation as to their functions. Foldseek search revealed that T6EC1 has a striking structural similarity to Colicin M (PDB: 4G76) (Fig. 3C, Purple). Colicin M targets lipid II, the peptidoglycan precursor, by hydrolyzing it in the periplasm, thereby affecting the cell wall (Chérier et al, 2021). We reasoned that since T6EC1 was toxic in the periplasm (Fig. 2D), and has strong structural similarity to Colicin M, then T6EC1 may have a similar effect on the cell wall. To explore this possibility, we expressed T6EC1 in the periplasm of E. coli BL21, and imaged the cells with fluorescence microscopy. We hypothesized that cells expressing T6EC1 would lose their characteristic rod shape and become round, due to loss of the protection of the cell wall against turgor pressure. By analyzing images of ~15,000 cells we found that, indeed, a high proportion of cells expressing T6EC1 in their periplasm were round (Fig. EV4A). As a control, we showed that E. coli BL21 co-expressing T6EC1 and the immunity protein T6IC1 kept their characteristic rod shape, via the automated analysis of ~22,000 cells (Fig. EV4A). Given these experiments, we propose that DUF3289 and DUF943 should be recognized as a putative toxin–antitoxin pair and re-annotated in domain databases (Interpro/Pfam) as toxin likely affecting cell wall and its cognate immunity protein, respectively. These results are in line with the hypothesis that T6EC1 affects the cell wall, perhaps similarly to colicin M. Further experiments are needed to confirm the details of the mechanism of action and to confirm that the putative effector is indeed secreted in a T6SS-dependent manner.

Sequence-based search for T6EC2, a member of Foldseek cluster 62, did not find any results useful in determining the function of the protein, but Foldseek search of Alphafold-predicted structures also revealed that T6EC2 had a hit to proteins with tox-REase-5

nuclease domains (EBI Alphafold db: AF-A0A2W7NTG7) (Fig. 3C). Taking these Foldseek hits into account, we hypothesized that T6EC2 degrades nucleic acids, leading to cell death. We imaged E. coli BL21 expressing T6EC2 in their cytoplasm after 1 h, but we did not see any aberration of the DAPI signal, as we had expected. We did, however, observe filamentation of cells suggestive of a severe cell division defect which may be because of the activity of T6EC2 (Fig. EV4B). In contrast, E. coli BL21 expressing empty pBAD and pET did not have a prevalence of these long, filamented cells (Fig. EV4B). Although this result does not conflict with the hypothesis, it is also not conclusive. We speculate that T6EC2 may be an RNAse, as other tox-REase-5-containing effectors were seen to degrade RNA (Yadav et al, 2021), and RNAse activity would not be visible with DAPI staining. Further in vitro studies may shed light on the specific mechanism of action of T6EC2.

Sequence-based search did not identify the possible function of T6EC3, a member of Foldseek cluster 275. The immunity protein T6IC3 has a predicted periplasmic localization signal, which is in line with the fact that the toxin has activity in the periplasm (Fig. 2D). Foldseek searches revealed a structural homology of the C-terminal domain of T6EC3 and SleB, a hydrolase that cleaves a specialized form of peptidoglycan during Bacillus spore germination (PDB: 4F55) (Fig. 3C) (Jing et al, 2012). Taken altogether, we hypothesized that T6EC3 targets the cell wall. We performed microscopy and expected that T6EC3 would cause cell rounding. E. coli BL21 expressing T6EC3 for thirty minutes showed many cells that had rounded (Fig. 4B). However, in contrast to the cell rounding seen that was caused by T6EC1, the rounding caused by T6EC3 expression had peculiar double-membrane structures, with DAPI signal in the inner membrane (Figs. 4B and EV4C). Given the wealth of different mechanisms bacteria use to disrupt the cell wall (Sibinelli-Sousa et al, 2021), we expected that other T6Es may act similarly. However, to the best of our knowledge, other studies that imaged T6Es that affect the cell wall showed cell rounding without separation of the membranes (Whitney et al, 2013; Russell et al, 2011; Wang et al, 2020), or elongated and/or "inflated" cells that burst (Sibinelli-Sousa et al, 2020; English et al, 2012), or simply bursting (Altindis et al, 2015). Using time-lapse imaging, we observed rod-like bacteria that lost their shape and became rounded, as well as rounded cells that eventually burst (Movie EV1). We speculate that this unique phenotype may reveal a new T6E cell wall degradation mechanism that biophysically leads to the separation of the inner and outer membranes. To further understand the mechanism of action, we compared the structure of T6EC3 and SleB. SleB has a catalytic glutamate residue (Li et al, 2012) that aligns in 3D space with a glutamate residue in T6EC3. We hypothesized that this analogous glutamate in T6EC3 is necessary for its activity. To test this, we performed a point mutagenesis of the catalytic glutamate to glutamine (E108Q), which makes a minimal change to the side-chain structure, yet changes its chemical nature. The mutant was not toxic to E. coli (Fig. 4D), suggesting that this glutamate is necessary for its mechanism of action, therefore supporting the idea that T6EC3 is indeed a lytic transglycosylase.

Foldseek searches of T6EC4, a member of Foldseek cluster 268, against the PDB database of experimentally-derived crystal structures only identified partial matches to very short segments of methyltransferases, which was largely uninformative. However, Foldseek search versus the Alphafold-predicted structure database

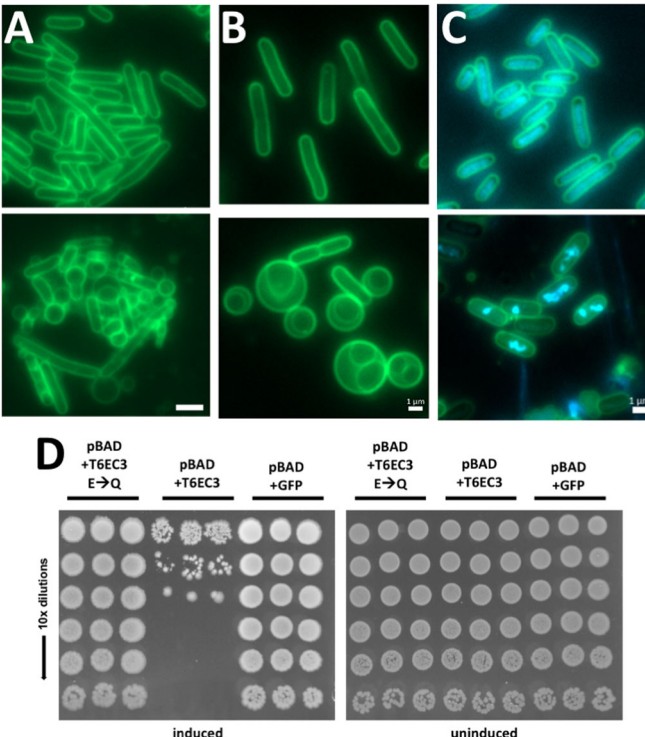

**Figure 4. Toxin molecular activities are consistent with hypotheses formulated by Foldseek.**

(A) FM1-43 stained membranes of *E. coli* BL21; top: expressing both T6EC1 and T6IC1; bottom: T6EC1 only (empty vector control). Imaging was performed after 1 h of induction of 0.01 mM IPTG and 0.2% arabinose. Scale bar = 2 μm. (B) top: *E. coli* BL21 expressing pBAD24 and pET29b empty vectors; bottom: pBAD24 with T6EC3 and pET29b with T6IC3 after 15 min of induction with 0.01 mM IPTG and 0.2% Arabinose. Note the immunity protein is expressed but does not cause full saving activity. Scale bar = 1 μm. (C) top: Empty vectors pBAD24 and pET29b, bottom: pBAD24 with T6EC4 and pET29b with T6IC4 were co-expressed in *E. coli* BL21 and imaged after 1 h of 0.2% arabinose induction. Note the immunity protein is expressed but does not cause full saving activity. Blue channel represents DAPI staining. Scale bar = 1 μm. (D) Drop assay showing *E. coli* BL21 expressing GFP (control), T6EC3, or T6EC3 E108Q mutation on pBAD24 plasmid vectors. Three columns of drops are biological replicates. The induction is carried out using 0.2% Arabinose. Source data are available online for this figure.

reveals a high structural similarity to Tox-REase-5 nuclease domains (Pfam15648), including to a known T6E from *Pseudomonas aeruginosa* PAO1, TseT (Burkinshaw et al, 2018) (Fig. 3C) (EBI Alphafold db: AF-Q9HXA6-F1-model_v4). A pairwise Blast search of the amino acid sequence of T6EC4 and TseT revealed no sequence similarity (Fig. EV4D), highlighting the benefit of structure-based searches in determining toxin function. We therefore hypothesized that T6EC4 is an endonuclease. We imaged *E. coli* BL21 with T6EC4 and T6IC4 plasmids, with induction of T6EC4 only. The T6EC4-only strain was highly toxic, so we therefore imaged the strain with the presence of T6IC4, which lessened the toxicity, even when the immunity protein is not induced (Fig. 4C). We stained the cells with DAPI to investigate the effect of T6EC4 on the nucleotides. After 1 h of toxin induction, we observed that the DAPI signal was compressed into puncta and irregular shapes in the cell, suggesting damage to the structural

integrity of the DNA (Fig. 4C). Although further in vitro experiments on purified protein must be performed to ultimately confirm the mechanism of action, this result is in line with the hypothesis that T6EC4 is an endonuclease.

These experiments show that hypotheses formulated based on the sensitive structure-based Foldseek search are supported by basic experimental results. Overall, our study shows the utility of the novel structural bioinformatic tools Alphafold2, Alphafold-Multimer, and Foldseek in functional gene annotation and in particular for mining genomes for T6E-immunity pairs. These tools help with more efficient clustering of putative T6Es, prediction of putative T6E-immunity protein–protein interactions, and in finding structural homologs at an unprecedented speed and scale, which assists in hypothesis building. We offer this study as a first proof-of-concept for the use of these structural bioinformatic tools in T6E discovery, as well as the list of classifications of the putative T6Es into 517 structural clusters, their predicted immunity proteins, and their manually curated annotations. We hope that these tools will be widely adopted for further applications in the T6SS field.

## Discussion

This study demonstrates the effectiveness of new structural bioinformatics tools in exploring genomic data to uncover T6E-immunity pairs. Structural clustering outperforms sequence-based methods, yielding 517 clusters in the Proteobacteria T6E structure space. Using Alphafold-Multimer's ipTM score, we predicted binding between T6Es and immunity genes, identifying candidate immunity proteins for 231 clusters. Foldseek facilitates fast and sensitive structure-based searches, enhancing Pfam-based annotations, and aiding hypothesis generation for possible mechanisms of action. Experimental validation confirmed the toxicity of four T6Es to *E. coli*, with their immunity proteins rescuing toxicity. We provide the first evidence that DUF3289 is a cell wall targeting colicin M homolog and DUF943 is its immunity protein. T6EC3 shares a catalytic glutamate residue with its homolog sleB that is essential for its toxicity. This proof-of-concept shows the utility of these novel structural bioinformatic tools, especially to experimental scientists (Ramola et al, 2022), for the discovery of novel T6E-immunity pairs.

We are aware that in order to define a gene as a bona fide T6E, one needs to provide evidence of T6SS-dependent secretion and/or translocation, and that the presence of the gene in the attacking strain needs to correlate with higher fitness in competition assays against a prey strain that lacks the cognate immunity protein. We emphasize that the main focus of this study was to explore the use of novel tools to see how they could augment T6E discovery and functional characterization. We were careful to note that Foldseek-based annotation is useful for generating accurate *hypotheses* of mechanisms of action. We sought to perform simple experiments to show that empirical data supports, rather than falsifies, our hypotheses. The experiments presented in this study are indeed in line with the hypotheses generated with the help of Foldseek, but are not meant to be exhaustive. Future work beyond the scope of this manuscript will explore each of these T6E structural clusters for a higher-resolution understanding of their activities and of the specific molecular determinants of their mechanisms of action.

It is becoming increasingly popular to apply algorithms to large genomic datasets to prediction of gene function, and specifically to

T6SS effectors and their immunity proteins. The methodology in this study can be expanded to other N-terminal T6SS markers like MIX, FIX, and RIX domains, amongst others (Kanarek et al, 2023; Salomon et al, 2014; Dar et al, 2018; Jana et al, 2019). In fact, this study provides a proof-of-concept that can be further expanded to other secretion system effectors, as well as any genes/proteins that follow a toxin–antitoxin schema, such as T5SS and T7SS effectors (Ulhuq et al, 2020). We suggest that calculating ipTM scores of putative T6E-immunity pairs can lower the risk of failed experiments and save precious laboratory resources. Furthermore, the elucidation of possible mechanisms of action by structural search raises the chances of designing more pinpointed downstream experiments rather than wasting time and resources on more speculative hypotheses. Indeed, such studies using Alphafold-Multimer for toxin–antitoxin studies (Ernits et al, 2023) and the use of structural clustering to bridge sequence gaps in pathogenic effectors (Seong and Krasileva, 2023) has already begun. Alphafold-Multimer can also be used to identify novel T6SS chaperones/adapters, since they bind effectors as well and are generally located upstream (Manera et al, 2022; Liang et al, 2015; Unterweger et al, 2015; Liu et al, 2020). With new tools, the same data can be observed from a new angle, bringing an exciting new era with a lot of promise.

We note that about half of the T6E clusters lack an immunity gene. There are many reasons for this, including being too conservative in assuming the immunity protein is always directly downstream, false negatives in the logistic regression model, poor folding by Alphafold-Multimer, more general immunity mechanisms like cell wall modification, immunity islands, and stress responses (Hersch et al, 2020a; Le et al, 2020; Ross et al, 2019; Hersch et al, 2020b), and because the effector may be anti-eukaryotic against a eukaryotic-specific target (obviating the need to protect against self-intoxication/sister cell injections). By noting these characteristics of a putative effector-immunity pair, we can better predict if they are antibacterial or anti-eukaryotic, even if we do not have a good hypothesis for a specific mechanism of action.

We also note that the C-termini in our 517 clusters are not always toxic effectors. As seen above, some were predicted to include TTR domains, which is presumably for interaction with cargo effectors, as this is its role in binding Tle1 (Flaugnatti et al, 2020). Although we cannot refute the possibility that TTR exerts toxicity by binding and inhibiting critical proteins, such as ribosome components. We also detected DUF4123 domains in the C-terminal clusters, which are chaperones of the T6SS, which also may be for assisting cargo effectors to be loaded onto the T6SS machinery. T6SS effectors can also be involved in scavenging metals (Lin et al, 2017). So although many of the C-terminal clusters are likely toxic effectors, as seen by their annotated enzymatic activities, there are some domains which play other roles.

T6EC2 did not have the DAPI puncta that we observed with T6EC4 expression in _E. coli_. Since T6EC2 has a tox-REase-5 domain, we compared it to another T6E called TseTBg, which also contains a tox-REase-5 domain. Interestingly, TseTBg has dual nuclease activity, meaning it can degrade both DNA and RNA, according to in vitro assays (Yadav et al, 2021). It is possible that our specific protein degrades RNA preferentially, and therefore we did not see any disturbances of DAPI staining. Further studies could shed more light on the exact mechanism of action of T6EC2.

The cause of the unique T6EC3 rounding phenotype where we saw membrane separation between inner and outer membrane is unclear. Further studies may elucidate the exact mechanism, but it is also important to note that it may be due to the conditions of our specific experimental setup. However, given the fact that its homolog SleB has a unique active site topology as compared to other lytic transglycosylases (Jing et al, 2012), we hypothesize that there is indeed a true phenomenon to explore deeper.

In conclusion, our study demonstrates the effectiveness of structural bioinformatic tools in discovering novel T6E-immunity pairs. Structural clustering identified 517 clusters, while Alphafold-Multimer helped predict 231 candidate immunity proteins (Dataset EV1). Foldseek search helped annotate 265 clusters (Dataset EV1). This study establishes a proof-of-concept for the role of these tools in advancing our understanding of T6E-immunity pairs.

# Methods

## Genomic pipeline

Using the IMG/M database (Chen et al, 2019, 2017; Markowitz et al, 2012), we defined a T6SS-encoding genome that contains at least eight T6SS marker domains (Dataset EV2). This database of T6SS-encoding genomes ($n = 17,920$) was virtually all made up of Proteobacteria, because the marker domains were largely defined by T6SS subtype i (see markers in Dataset EV2), excluding the subtype ii and iii T6SS from this study. From these genomes, we extracted genes containing Hcp, VgrG, PAAR, and PAAR-like Pfam domains, which include genes with "core" (structure only) T6SS domains, as well as those that match the architecture of specialized T6Es (i.e., those with a C-terminal extension), with a total of 203,608 protein sequences. To remove redundancy, we used MMseqs2 (version 14.7e284). Specifically, we used mmseqs cluster, with conservative parameters of 50% identity over 90% coverage (over both query and subject), to only remove true redundancy, resulting in 10,778 protein sequences. A significant number were found to be partial genes because they were encoded on the end of a scaffold, so we removed those, resulting in 6535 protein sequences. We filtered for those sequences with C-terminal extensions that are large enough to encode a domain, with boundaries defined by the end coordinates of the alignments to core T6SS domains in each gene. For PAAR and Hcp, we filtered for C-terminal domains with >=120 amino acids after the end of the PAAR or Hcp domain. Because VgrG's domains did not reach the end of the core structural domain we first took those with an overall length greater than >796 amino acids, based on the mean length of the core VgrG (no C-terminal extension) distribution (724 aa) and added one standard deviation (72 aa). Overall this resulted in 1315 protein sequences. However, based on this simple cutoff, we found that some of the C-termini of the VgrG still had, upon manual inspection of the structures, extra long DUF2345-like needle domains remaining, which are structural in nature, and did not constitute specialized effectors. We used these needle domains to redefine the end coordinate of VgrG, and removed those with <120 amino acid extensions. Final coordinate values for cutoff are shown in Supplementary Data 1. This resulted in 1192 Hcp, VgrG, and PAAR that were considered possible specialized effectors (with long

C-terminal extensions). We then clustered the C-terminal domains using MMseqs2 as above, except with 50% identity over 85% coverage, resulting in 1065 C-terminal clusters. After filtering for this final sequence dataset, this overall represents 21,939 separate proteins in the IMG/M database.

## Protein structure prediction

Protein structure prediction was carried out using localcolabfold (Mirdita et al, 2022) version 1.5.0 with model-type AlphaFold2-multimer-v3 (Evans et al, 2022; Jumper et al, 2021; Jumper and Hassabis, 2022), num-recycle 20, stop-at-score 90, and rank multimer. The best model was used in downstream analyses (i.e., the rank_001 model). Effector-immunity pairs were folded together, i.e., as one protein sequence with an asterisk separating them.

## Structural clustering

Structural clustering was carried out using Foldseek (version 7.04e0ec8), with the cluster subcommand, cov_mode 0 (coverage over both representative and subject), coverage 0.85, and $P$ value of 1e-6. This resulted in 517 clusters representing the 1065 input structures.

## Immunity protein prediction model

Experimentally validated T6E-immunity pairs from SecReT6 database (Li et al, 2015; Zhang et al, 2023) was accessed September 2022. Cognate T6E-immunity pairs and shuffled (randomized pairings of T6Es and noncognate immunity proteins) had their protein structure predicted with localcolabfold with the settings mentioned above. ipTM scores, pDockQ scores, or mean intermolecular PAE scores (or combinations of those) were used to train a logistic regression model (implemented with sklearn.linear_model.LogisticRegression (Pedregosa et al, 2011), C = inverse of regularization strength = 0.1) that predicted if the binding was "expected" (cognate pairs) or not expected (randomized pairs). Only those that were well folded overall (pTM score >= 0.5) were used to train the model. In total, 70% of the data were used for training, and 30% of the data for testing.

Data for finding novel immunity proteins was found by taking genes that were small (<700 amino acids; the largest experimentally tested immunity protein from SecReT6 = 699 amino acids), and directly downstream of the putative T6Es (<=100 nucleotides), and facing in the same direction as the putative T6E.

The ipTM score was mainly used for evaluating the probability of binding, as this metric was used successfully for other protein–protein interaction studies (Yin et al, 2022; Teufel et al, 2023; Evans et al, 2022). ipTM is a multimer extension of the pTM score from Alphafold2 (Jumper et al, 2021). The ipTM is listed in the output JSON file. We also compared two other methods, pDockQ (Bryant et al, 2022a) and also our own method based on the PAE score. For our own method, we took the output JSON file containing the pLDDT and PAE scores were processed first to find continuous tracts of pLDDT scores above with a rolling average above 70 (window size = 30). This defines the start coordinates and end coordinates of the protein with trustworthy folding. Then we retrieved the intermolecular PAE scores corresponding to these

start and end coordinates, which define rectangular matrices in the bottom left and top right quadrants of the PAE plot. The PAE score presented in the figures of this study is derived by normalizing and reversing the raw PAE score by applying the formula: $(30 - \text{pae})/30$, making the highest confidence scores 1, and the lowest confidence 0. Since sometimes multiple continuous tracts of pLDDT scores with a rolling average above 70 were present (i.e., multiple domains of high quality folding), we evaluated all possible combinations of intermolecular interactions, and took the domains with the maximum score. Those proteins with poor folding (no continuous tracts of pLDDT score above 70) returned no score.

## Structural search

PDB files produced by AlphaFold2 modeling were uploaded onto the Foldseek search server (van Kempen et al, 2024) (https://search.foldseek.com/), with search for Alphafold/Uniprot50 v4, Alphafold/Swiss-Prot v4, Alphafold/Proteome v4, and PDB 20231120. Foldseek search using the command line interface was also carried out against the Foldseek Uniprot database v4.

## Protein annotation

Pfam annotation was run with fasta sequences as input to hmmscan, part of HMMER 3.3.2. The Pfam HMM profile database (Bateman et al, 2004; Finn et al, 2008; El-Gebali et al, 2019) is listed as updated October 2021. Results were filtered for $E$-value < 1e-3 and HMM coverage of over 0.7.

Foldseek web interface was used to manually view alignments, and more information regarding annotation was taken from the Uniprot (UniProt Consortium, 2019) database.

## Drop assay

Putative T6Es were synthesized (codon optimized for *E. coli*) and cloned into plasmids (either pET29 for immunity genes, or pBAD24 for toxin genes) by Twist Bioscience (California, USA). T6EC1, T6IC1, and T6EC3 had TAT sequences from *E. coli* trimethylamine N-oxide reductase (torA) added to their N-terminal for periplasmic localization (T6IC3 already has a natural periplasmic localization signal sequence). Using heat shock, pET29 + immunity genes were transformed into *Escherichia coli* BL21. Transformed colonies were then used to create competent cells, which were in turn transformed with pBAD24 + toxin genes. Overnight cultures of the strains harboring the vectors of interest were grown in LB containing kanamycin and ampicillin. Cultures were normalized to 0.5 $OD_{600}$ and subsequently serially ten-fold diluted. Dilutions were spotted on LB agar containing the proper selection and inducer; plates with either 1% Glucose and 0.01 mM IPTG, or 0.2% Arabinose and 0.01 mM IPTG. Blinding was not employed as it was deemed not relevant to the experimental design and objective. A Dunn test was performed between the "T6E and empty, induced" versus the "T6E and immunity, induced" groups using the R library Dunn.test.

## Microscopy and Image quantification

Freshly transformed colonies of *E. coli* BL21 with pBAD24 + toxin gene, with or without pET29 + immunity gene were grown

overnight in liquid cultures of LB supplemented with the appropriate antibiotic. Overnight cultures were diluted 1:100 and grown for 3 h until the cells were in the log phase. Induction of expression of either toxin, toxin and immunity, or immunity only was performed where relevant. After incubation, 200 µL of cells were centrifuged at 10,000×g, and the pellet was resuspended in 5 µl of PBS with 1 mg/ml membrane stain FM1-43 (Thermo Fisher Scientific T35356) and 2 µg/ml DNA stain 4,6-diamidino2-phenylindole (DAPI; Sigma-Aldrich D9542-5MG). Microscope slides or agarose pads (for time-lapse; 1% agarose) were imaged on an Axioplan2 inverted fluorescence microscope on ZEN Blue software version 3.1 (Zeiss).

ZEN Blue or ZEN lite software (Zeiss) was used to process and export images for figures. Image quantification for some images was carried out using a deep learning package developed by our lab (https://github.com/noamblum/cellstats), which is based on human-in-the-loop training of the foundation model Cellpose (Stringer et al, 2021) using microscopy images from our laboratory.

### Visualization

Proteins were visualized using ChimeraX (Pettersen et al, 2021). Some figures were created using BioRender (BioRender.com).

## Data availability

The computer code produced in this study is available in the following database: Modeling computer scripts: GitHub (https://github.com/alexlevylab/T6E_discovery).

The source data of this paper are collected in the following database record: biostudies:S-SCDT-10_1038-S44320-024-00035-8.

## Peer review information

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

## Acknowledgements

The authors thank Yaara Oppenheimer-Shaanan for technical help with microscopy. AL is generously supported by the Israeli Science Foundation (Grants #1535/20, #3300/20), the Israeli Ministry of Agriculture, and the Volkswagen Foundation (ZN4041). AMG is generously supported by the Kaete

Klausner Scholarship and was supported by a scholarship from the Israeli Ministry of Aliyah and Integration.

## Author contributions

**Alexander M Geller**: Conceptualization; Resources; Data curation; Software; Formal analysis; Supervision; Funding acquisition; Validation; Investigation; Visualization; Methodology; Writing—original draft; Project administration; Writing—review and editing. **Maor Shalom**: Formal analysis; Investigation; Visualization; Methodology. **David Zlotkin**: Conceptualization; Data curation; Software; Formal analysis; Investigation; Visualization; Methodology. **Noam Blum**: Conceptualization; Data curation; Software; Methodology. **Asaf Levy**: Conceptualization; Supervision; Funding acquisition; Visualization; Writing— original draft; Project administration; Writing—review and editing.

Source data underlying figure panels in this paper may have individual authorship assigned. Where available, figure panel/source data authorship is listed in the following database record: biostudies:S-SCDT-10_1038-S44320-024-00035-8.

## Disclosure and competing interests statement

The authors declare no competing interests.

# Expanded View Figures

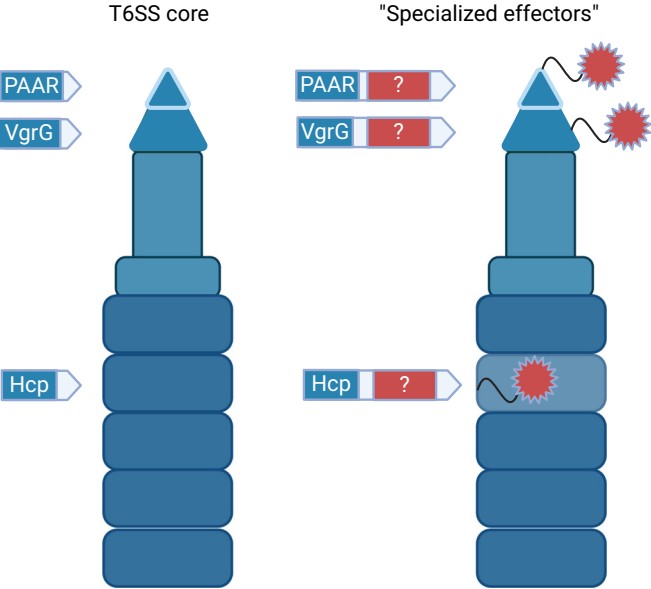

**Figure EV1. T6SS structure and specialized effectors.**

The T6SS is a contractile injection system made up of multiple components, including a donut-shaped tube component from hexamers of Hcp, and a sharp tip component made of VgrG and PAAR; we call these core T6SS components (right). Sometimes, these components have long C-terminal extensions that encode protein domains with T6SS effectors (left, red). Structurally, this results in what are called "specialized effector" proteins (sometimes referred to as "evolved" effectors). The C-terminal extensions lead to the loading of the C-terminal effector domain onto the T6SS, simply due to the fact that it is covalently attached to N-terminal core T6SS domains. The genetic organization of an N-terminal core domain and a C-terminal effector domain provides a characteristic genetic signature making specialized T6Es easily identifiable in large genomic databases.

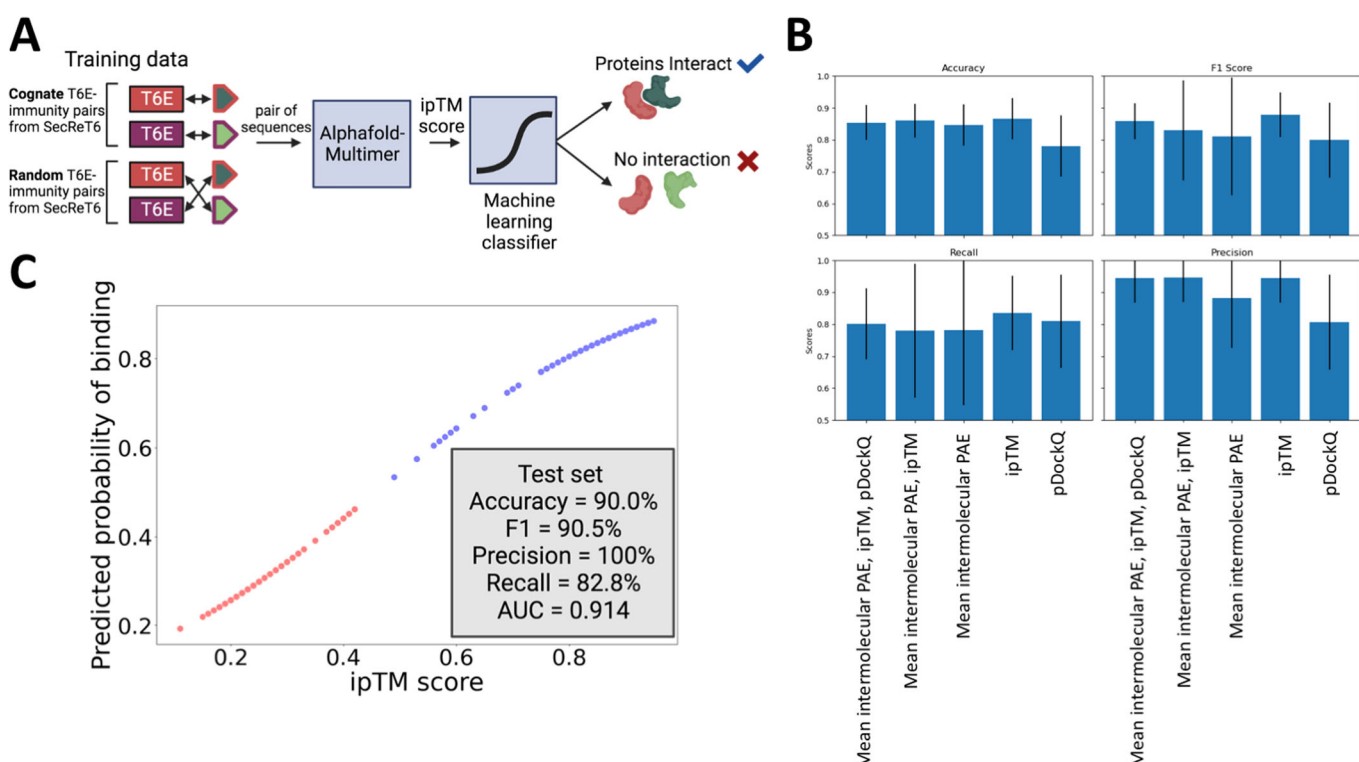

**Figure EV2. Training a logistic regression model for classification of T6E-immunity interaction.**

(A) Pipeline for training logistic regression model. T6E-immunity pairs from the SecReT6 database (Zhang et al, 2023; Li et al, 2015) were used as positive set input to Alphafold-Multimer, while shuffled (randomized) pairs of noncognate T6Es and immunity proteins were used as negative set inputs. The model was trained on ipTM score, a default output of Alphafold-Multimer. (B) Various iterations of the model were built, including based on multiple inputs beyond ipTM, like Mean intermolecular PAE, and pDockQ, and combinations of these inputs. ipTM alone was chosen in the end because it alone equaled or outperformed the other inputs in terms of accuracy, precision, F1 score, and recall. Bars represent mean, error bars represent standard deviation of $k = 10$ instances of $k$-fold validation performed on a dataset of $n = 164$ pairs (95 in positive set, 69 in the negative set). (C) Trained model evaluated on both training and test set reveals the shape of the trained logistic regression curve. Blue dots are predicted protein interactions, and red are predicted non-interactions. Inset shows test set accuracy, F1, precision, recall, and AUC values.

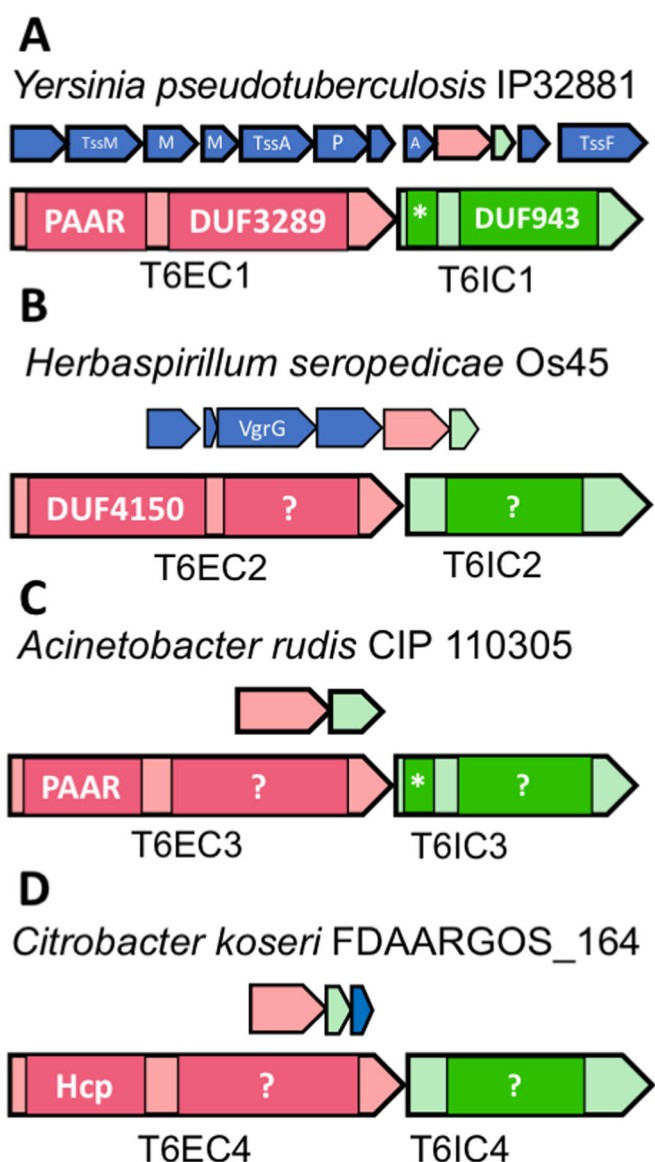

**Figure EV3. T6EC1-4 and T6IC1-4 are putative T6E-immunity pairs.**

Genes in operons are shown on top, and protein/domain architecture is shown on the bottom of each panel. (**A**) T6EC1 and T6IC1 are encoded in a T6SS operon (top; M = TssM, P = PAAR, A = TssA). T6IC1 has an N-terminal transmembrane domain signal, marked with an asterisk. The transmembrane prediction is as follows: residues 1–6 are inside, residues 7–26 are transmembrane helices, and residues 27–157 are outside (i.e., in the periplasm). (**B**) T6EC2 and T6IC2 are in an orphan/auxiliary operon containing a VgrG gene. (**C**) T6IC3 has an N-terminal signal sequence strongly suggesting it is localized to the periplasm. (**D**) T6EC4 and T6IC4 are in an orphan/auxiliary operon.

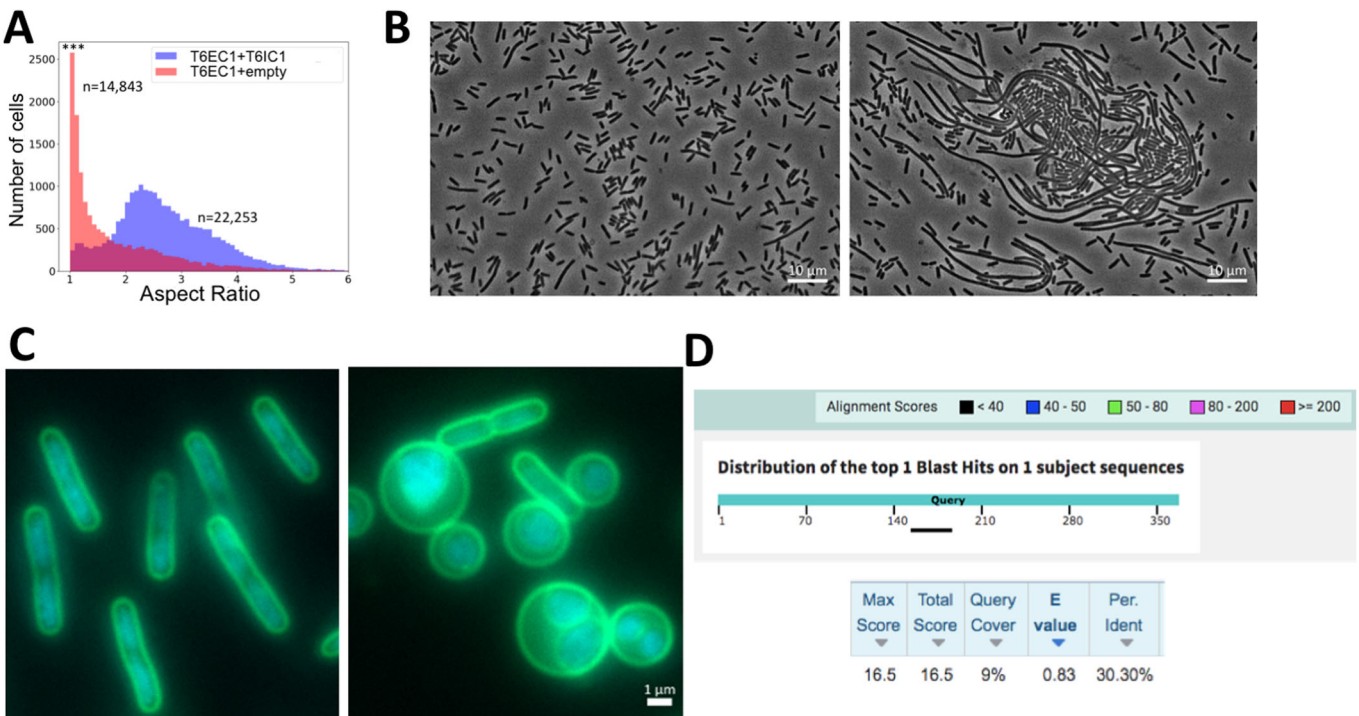

**Figure EV4. Experimental exploration of T6EC functions.**

(A) Histogram of aspect ratio (a measure of length/width). Blue = both T6EC1 and T6IC1, Red = T6EC1 only, purple = overlap of histograms. An aspect ratio of 1 is a circle (equal length and width). Asterisks indicate statistical significance (Mann–Whitney $U = 258904636$, $P = 1E-1871$). (B) Expression of T6EC2 results in filamentation. *E. coli* with an empty pBAD24 and pET29b vectors (left) or pBAD24 with T6EC2 and an empty pET29b vector (right) was imaged after 1 h of incubation with IPTG and arabinose. (C) *E. coli* BL21 expressing pBAD24 and pET29b empty vectors (left) or pBAD24 with T6EC3 and pET29b with T6IC3 after 15 min of induction with 0.01 mM IPTG and 0.2% Arabinose (right). Note the immunity protein is expressed but does not cause full saving activity. The same cells are shown in Fig. 4B, but here, the signal is an overlay of membrane and DNA signals from FM1-43 and DAPI, respectively. (D) Sequence-sequence search by BLAST of T6EC4 and TseT from *Pseudomonas aeruginosa* PAO1 has no significant similarity. Source data are available online for this figure.

