## [Peer Review File · Molecular Systems Biology]

Identification of type VI secretion system effector-immunity pairs using structural bioinformatics

Alexander Geller, Maor Shalom, David Zlotkin, Noam Blum, and Asaf Levy

Corresponding author(s): Asaf Levy (alevy@mail.huji.ac.il)

Review Timeline:

Submission Date:	2nd Jan 24
Editorial Decision:	23rd Feb 24
Revision Received:	4th Mar 24
Editorial Decision:	19th Mar 24
Revision Received:	24th Mar 24
Accepted:	9th Apr 24

Editor: Poonam Bheda

Transaction Report:

23rd Feb 2024

Manuscript Number: MSB-2024-12200

Title: Identification of type VI secretion system effector-immunity pairs using structural bioinformatics

Dear Dr Levy,

Thank you again for submitting your work to Molecular Systems Biology. We have now heard back from the three reviewers who agreed to evaluate your study. As you will see below, the reviewers appreciate that the proposed approach addresses a timely topic. However, they raise a series of concerns, which we would ask you to address in a major revision.

I think that the recommendations of the reviewers are rather clear and I therefore do not see the need to repeat the comments listed below; however, editorially we would like to reiterate the point from Reviewer 3 that the analysis codes should be provided in a publicly accessible repository such as Github.

All other issues raised would need to be satisfactorily addressed. Please let me know in case you would like to discuss in further detail any of the comments, I would be happy to schedule a call.

We require:

1) A .docx formatted version of the manuscript text (including legends for main figures, EV figures and tables). Please make sure that the changes are highlighted to be clearly visible. Alternatively you may choose to submit your manuscript as a LaTeX file.

4) A .docx formatted letter INCLUDING the reviewers' reports and your detailed point-by-point responses to their comments. As part of the EMBO Press transparent editorial process, the point-by-point response is part of the Peer Review File (PRF), which will be published alongside your paper.

5) A complete author checklist, which you can download from our author guidelines (<https://www.embopress.org/page/journal/17574684/authorguide#submissionofrevisions>). Please insert information in the checklist that is also reflected in the manuscript. The completed author checklist will also be part of the PRF.

6) Please note that all corresponding authors are required to supply an ORCID ID for their name upon submission of a revised manuscript.

7) It is mandatory to include a 'Data Availability' section after the Materials and Methods. Before submitting your revision, primary datasets produced in this study need to be deposited in an appropriate public database, and the accession numbers and database listed under 'Data Availability'. Please remember to provide a reviewer password if the datasets are not yet public (see <https://www.embopress.org/page/journal/17574684/authorguide#dataavailability>).

In case you have no data that requires deposition in a public database, please state so in this section. Note that the Data Availability Section is restricted to new primary data that are part of this study. This study includes no data deposited in external repositories.

8) For data quantification: please specify the name of the statistical test used to generate error bars and P values, the number (n) of independent experiments (specify technical or biological replicates) underlying each data point and the test used to calculate p-values in each figure legend. The figure legends should contain a basic description of n, P and the test applied. Graphs must include a description of the bars and the error bars (s.d., s.e.m.). Please provide exact p values.

9) Our journal encourages inclusion of *data citations in the reference list* to directly cite datasets that were re-used and obtained from public databases. Data citations in the article text are distinct from normal bibliographical citations and should directly link to the database records from which the data can be accessed. In the main text, data citations are formatted as

follows: "Data ref: Smith et al, 2001" or "Data ref: NCBI Sequence Read Archive PRJNA342805, 2017". In the Reference list, data citations must be labeled with "[DATASET]". A data reference must provide the database name, accession number/identifiers and a resolvable link to the landing page from which the data can be accessed at the end of the reference. Further instructions are available at .

<https://www.embopress.org/page/journal/17574684/authorguide#expandedview>

11) For more information: There is space at the end of each article to list relevant web links for further consultation by our readers. Could you identify some relevant ones and provide such information as well? Some examples are patient associations, relevant databases, OMIM/proteins/genes links, author's websites, etc...

12) Author contributions: CRediT has replaced the traditional author contributions section because it offers a systematic machine readable author contributions format that allows for more effective research assessment. Please remove the Authors Contributions from the manuscript and use the free text boxes beneath each contributing author's name in our system to add specific details on the author's contribution. More information is available in our guide to authors.

13) Disclosure statement and competing interests: We updated our journal's competing interests policy in January 2022 and request authors to consider both actual and perceived competing interests. Please review the policy <https://www.embopress.org/competing-interests> and update your competing interests if necessary.

14) Every published paper now includes a 'Synopsis' to further enhance discoverability. Synopses are displayed on the journal webpage and are freely accessible to all readers. They include a short stand first (maximum of 300 characters, including space) as well as 2-5 one-sentences bullet points that summarizes the paper. Please write the bullet points to summarize the key NEW findings. They should be designed to be complementary to the abstract - i.e. not repeat the same text. We encourage inclusion of key acronyms and quantitative information (maximum of 30 words / bullet point). Please use the passive voice. Please attach these in a separate file or send them by email, we will incorporate them accordingly.

Please also suggest a striking image or visual abstract to illustrate your article as a PNG file 550 px wide x 300-600 px high. Share synopsis text and image, as well as eTOC:

Please note that these would be the final versions and changes during proofing are usually not allowed

15) As part of the EMBO Publications transparent editorial process initiative (see our policy here: https://www.embopress.org/transparent-process#Review_Process), Molecular Systems Biology will publish online a Peer Review File (PRF) to accompany accepted manuscripts.

In the event of acceptance, this file will be published in conjunction with your paper and will include the anonymous referee reports, your point-by-point response and all pertinent correspondence relating to the manuscript. Let us know whether you agree with the publication of the PRF and as here, if you want to remove or not any figures from it prior to publication.

Please note that the Authors checklist will be published at the end of the PRF.

Molecular Systems Biology has a "scooping protection" policy, whereby similar findings that are published by others during review or revision are not a criterion for rejection. Should you decide to submit a revised version, I do ask that you get in touch after three months if you have not completed it, to update us on the status.

I look forward to receiving your revised manuscript.

Yours sincerely,

Poonam Bheda

Poonam Bheda, PhD

Use the link below to submit your revision:

Reviewer #1:

This manuscript describes a timely work that uses novel and powerful predicted tools (A.I.) that allow the advance of the T6SS field. Specifically, the authors applied structural bioinformatic tools to identify and annotate 517 putative T6SS effector families from a dataset of 17,920 bacterial genomes. Through structural clustering and a logistic regression model, they also identified candidate immunity proteins for 231 T6E families and validated four novel EI pairs through experiments in *E. coli*. The novelty of the work resides in the use of structural clustering instead of sequence-based clustering, increasing the number of putative EI families identified; and structure-based annotation increasing the quality of the functional annotations.

I have some minor comments:

General comments

The authors have used evolved structural components such as Hcp, PAAR, PAAR-like and VgrG for the identification of the putative effectors. I agree that this approach is appropriate, but I am curious to know why they have not used other classical T6SS markers like MIX, FIX, RIX or even adaptors like EagR or Tap for this search, or the classical structural non-evolved components. All of them would have been useful to increase the reach of the study.

I would like to kindly suggest the improvement of the manuscript's English language. Although the overall content is well-structured and clear, some sentences appear to be grammatically complex or could benefit from smoother transitions.

The content of the Discussion section has many redundancies from the Results section. A more distinct separation between the two sections, with the Discussion focussing on the interpretation and implications of the results rather than restating them, would improve the readability of the manuscript.

The authors emphasise that the manuscript serves as a valuable resource for the scientific community by offering predictions for 265 T6E domain families. However, the presentation and explanation of the supplementary data make it challenging for users to effectively use this information. The manuscript should include a more detailed and accessible explanation of the supplementary data to ensure clarity and ease of use for the scientific community, increasing the impact of this work.

Specific comments

Line 50- it is not only sometimes that they are called "cargo" effectors, it is how the field has defined them.

Line 55 - the word "effector" is repeated.

Line 62. Immunity proteins also protect from sister cells firing and not only self-intoxication.

Line 67. Type VI effector-immunity pairs can be denoted as EI pair.

Line 71. Authors could consider including here an Opinion article discussing this subject (PMID: 33687778 DOI: 10.1111/1462-2920.15457)

Line 141. The authors could consider specifying that DUIF4150 is a PAAR-like domain.

Figure 2 D. The Y-axis is lacking a clear title indicating the represented parameter (survival of *E. coli* BL21).

Line 262-264 The authors suggest that because the T6IC1 putative immunity gene is predicted to encode a transmembrane anchoring domain, the activity of T6EC1 could be in the periplasm. This is not necessarily true, especially if it does not have a signal to be transported to this compartment. I understand that the results confirm their hypothesis, but since they do not discuss the effect of the effector being expressed in the cytosol, this data is not conclusive. The target could be the side of the inner membrane that is facing the cytosol.

The Integrated Microbial Genomes and Microbiomes should be named IMG/M as indicated on their website and not IMG.

Lines 341-342 are repetitive to lines 342-344.

Line 403, the reference to Figure 4D is missing.

Reviewer #2:

The manuscript "Identification of type VI secretion system effector-immunity pairs using structural bioinformatics" from Geller et al. is interesting and pleasant to read. The work presents an interesting approach to predict protein-protein interactions in bacteria based on AlphaFold2 and FoldSeek, which are newborn technologies in the structural bioinformatics field. In particular, here the authors focus on the proteins belonging to the T6SS, an important bacterial machinery able to inject effectors into preys. The manuscript describes an in silico approach to detect bacterial proteins secreted by the T6SS and the associated immunity proteins that protect the bacteria from these proteins. The method allows functional annotation of effectors and detection of the immunity partner, and finally the database is shared with the community. Some selected predictions are experimentally validated. The authors focus on 'specialized' effectors, which are fused with components of the T6SS. Using sequence analysis, they extensively identify the specialized effectors in bacterial genomes. Structures of extra-domain were predicted with AlphaFold2 and the structural comparison with Foldseek was carried out to suggest functional annotations. Immunity proteins were predicted based on their genomic location (downstream and in the vicinity of effectors) and the AlphaFold prediction of the effector/immunity protein dimer. Experimental validation was carried out on four pairs of effector + immunity proteins.

The main ideas and hypotheses are clearly stated. The results support the claim. The findings will be of interest for the community of microbiologists. Here I list comments to improve the methods.

A possible validation of the classifier that they developed is to see how it behaves against noise. Thus I wonder what happens if you take all your known effector-immunity pairs and randomize them (ie, you randomize the pairing between the effector and immunity protein), and see what is the resulting iPTM score for the randomized pairs, and how they are classified. I guess iPTM should be low, but it is worth seeing to have additional trust on the classifier. Furthermore, it would be nice to see the same analysis for a set of predicted effector-immunity pairs, ie, randomize the pairs as described above.

The structural comparison with FoldSeek is used to propose a functional annotation of the specialized effector extra domains. At the level of structural similarity, one can expect that a given structure can match with several proteins of different functions. It would be interesting to know how this multiplicity is handled.

The authors performed a mutation on a candidate effector, and state that "we performed a point mutagenesis of the catalytic glutamate to glutamine, causing minimal changes in steric structures". Although it seems safe to postulate that the mutation will not perturb the structure, I would suggest the authors to be less affirmative.

Minor points

line 79 I think it is generally accepted to say that AF2 didn't solve the protein folding problem, but improved protein structure prediction, only for folded domains. Protein folding is a more complex problem involving the dynamics and the pathway of the process, which is still not systematically resolved.

line 97 thier -> their

The above evaluation is about the in silico approach, and does not concern the experimental validation.

Reviewer #3:

In this manuscript, the authors present a large scale integrated bioinformatic predictive structural analysis of effector and immunity genes of the type VI secretion system of the phylum Pseudomonodota. This pipeline leverages validated effector and immunity gene pairs to discover new pairs. Further, the authors validate the toxicity of 4 new previously unexplored effectors and demonstrate the neutralization capacity of the associated immunity genes. All together, the paper is well written and interesting, the method is powerful and provides more comprehensive insight into the spectrum of T6SS genes. Generally, I like the paper and I only have a few comments, which I list below.

My major comment is that I request that the authors significantly tone down the strength of language of their interpretations throughout the entire manuscript. The alphafold modeling, while powerful, is prediction only and should not be overinterpreted. For example, the RMSD values of the T6E1-4 effectors is somewhat poor. While these values reflect structural similarity across the whole proteins, he authors do not specifically compare in this structural analysis potential enzymatic active sites (e.g. colicin M <https://doi.org/10.1074/jbc.M109.093583>) and, lacking this comparison and followup experimental mutational analysis of potential active sites in T6E1-4, I think the authors need to make abundantly clear that effector mechanism is unknown and therefore speculation about mechanism should be clearly conveyed as it is - speculation.

The authors search for "core" proteins (Paar, vgrg, hcp, etc) but not for structural T6SS proteins. It seems possible that many genomes lack functional T6SS and I would think that assessing if the entire repertoire of T6SS structural proteins is found encoded in the genomes examined would be very important. Just as an example, in Lines 504-513, the authors discuss effectors

that lack immunity, which is counter to expectations. In these genomes, are structural genes missing? Another possibility, unless I misunderstand the analysis described on lines 239-243 in which I think only looked for immunity downstream of the effector, is that the immunity gene is actually upstream of the effector

The pie chart figure in 3B is unreadable. I recommend grouping/collapsing some of the categories into "other".

There is no information on biological replicates performed for the toxicity experiments and no description of statistical tests used - the authors must add this information. Likewise, there is no quantitative information regarding data from microscopy experiments. There are many useful tools (e.g. *microbeJ*) that facilitate quantitative cell biological examination of microscopy datasets and I urge the authors to add this, otherwise the examples shown lack value.

There is a tendency to cite recent reviews instead of primary literature and I encourage the authors to cite both. For example, the statement on line 62 regarding the role of immunity should include reference Hood et al PMID 20114026.

The authors should provide analysis code in a publicly accessible repository e.g. github.

Reviewer #1:

This manuscript describes a timely work that uses novel and powerful predicted tools (A.I.) that allow the advance of the T6SS field. Specifically, the authors applied structural bioinformatic tools to identify and annotate 517 putative T6SS effector families from a dataset of 17,920 bacterial genomes. Through structural clustering and a logistic regression model, they also identified candidate immunity proteins for 231 T6E families and validated four novel EI pairs through experiments in *E. coli*. The novelty of the work resides in the use of structural clustering instead of sequence-based clustering, increasing the number of putative EI families identified; and structure-based annotation increasing the quality of the functional annotations.

Thanks for your close reading of the paper and the detailed review. I am glad to hear you think it is timely and that you appreciate the novelty in using structure-based clustering.

I have some minor comments:

General comments

The authors have used evolved structural components such as Hcp, PAAR, PAAR-like and VgrG for the identification of the putative effectors. I agree that this approach is appropriate, but I am curious to know why they have not used other classical T6SS markers like MIX, FIX, RIX or even adaptors like EagR or Tap for this search, or the classical structural non-evolved components. All of them would have been useful to increase the reach of the study.

Since I wanted to focus and highlight the bioinformatic aspect, especially on the application of novel *in silico* tools, I wanted to make sure that I was 100% certain that the multitude of proteins under analysis were *bona fide* T6SS effectors. To ensure this, I used T6SS structural N-termini only. It is a great suggestion to apply the same methodology to any classical T6SS markers appearing at the N-terminal, as well as chaperones/accessories of the T6SS. Future research can now reliably use our pipeline and apply it as you suggest, which could lead to a more comprehensive database. I added a sentence about this in the Discussion section: "The methodology in this study can be expanded to other N-terminal T6SS markers like MIX, FIX, and RIX domains, amongst others."

I would like to kindly suggest the improvement of the manuscript's English language. Although the overall content is well-structured and clear, some sentences appear to be grammatically complex or could benefit from smoother transitions.

Okay thanks for the suggestion. I did notice in my fresh reading of the text that there were many run-on and complex sentences. I have gone through the manuscript and tried to simplify and complex sentences by splitting them up, and tried to add transitions where necessary.

For example, I cut a giant run on sentence to two sentences: "T6E-immunity pairs are actively researched because the T6SS is important to microbial ecology via their key role in niche colonization and pathogen-host interaction. **Furthermore**, T6Es can also be developed as

potential new antimicrobials, for medical and agricultural applications.” I found more examples of long sentences and did the same. Furthermore, I changed the tenses of the sentences to make them more active and less wordy.

I hope now you find it more readable, yet without any loss of understanding.

The content of the Discussion section has many redundancies from the Results section. A more distinct separation between the two sections, with the Discussion focussing on the interpretation and implications of the results rather than restating them, would improve the readability of the manuscript.

I changed the discussion to be more focused on the interpretation of the results rather than with restating the findings. I also removed Discussion topics from the Results section.

For example, I removed “Another benefit of *in silico* quantification of the binding affinity of putative T6E-immunity pairs is that it can aid in deciding which pairs to study further in the laboratory” from the results section, along with similar statements, which truly belong in the discussion section.

I also shortened the Discussion section’s first paragraph, which was a long 20 lines that detailed and restated the findings of the paper to about half size (12 lines), which is as minimal as I could get it. This should keep the focus on interpretation and further implications.

The authors emphasise that the manuscript serves as a valuable resource for the scientific community by offering predictions for 265 T6E domain families. However, the presentation and explanation of the supplementary data make it challenging for users to effectively use this information. The manuscript should include a more detailed and accessible explanation of the supplementary data to ensure clarity and ease of use for the scientific community, increasing the impact of this work.

This is really useful feedback because I do think the structural clusters could help a lot of researchers trying to formulate a mechanism of action of an unannotated effector protein, so it’s of paramount importance that it is understandable. I added a tab to the supplementary table with a detailed explanation of the clusters with a color-coded example going from individual gene accession IDs to structural clusters. Furthermore, I added a second tab explaining the “Immunity Protein” and “Annotation” tabs in more detail. I hope this helps to make it clearer.

Specific comments

Line 50- it is not only sometimes that they are called "cargo" effectors, it is how the field has defined them.

Fixed.

Line 55 - the word "effector" is repeated.

Fixed.

Line 62. Immunity proteins also protect from sister cells firing and not only self-intoxication. I added this information on line 62 (and on line 511).

Line 67. Type VI effector-immunity pairs can be denoted as EI pair.

I appreciate the suggestion, and it is true that I use “T6E-immunity pairs” many times in this manuscript, rather than the more typical “E-I pair” or “EI pair” nomenclature. The reason why I chose the former is because I also speak about the putative effectors alone many times. I felt that then the logical shorthand for the effectors would simply be “E”, and that does not sound good to me. So since it is a minor suggestion, I respectfully will keep our current nomenclature.

Line 71. Authors could consider including here an Opinion article discussing this subject (PMID: 33687778 DOI: 10.1111/1462-2920.15457)

Thanks, added.

Line 141. The authors could consider specifying that DUIF4150 is a PAAR-like domain.

Added this information in line 141.

Figure 2 D. The Y-axis is lacking a clear title indicating the represented parameter (survival of E. coli BL21).

I added this information to the Y axis as suggested.

Line 262-264 The authors suggest that because the T6IC1 putative immunity gene is predicted to encode a transmembrane anchoring domain, the activity of T6EC1 could be in the periplasm. This is not necessarily true, especially if it does not have a signal to be transported to this compartment. I understand that the results confirm their hypothesis, but since they do not discuss the effect of the effector being expressed in the cytosol, this data is not conclusive. The target could be the side of the inner membrane that is facing the cytosol.

This is a great point, and based on what is written, you are right that multiple models of possible subcellular localization can be formulated. I accidentally forgot to include more details about the transmembrane helix in the text that explain why I chose the periplasmic hypothesis. I forgot to write that the transmembrane helix is actually predicted to be largely facing outside (to the periplasm), so I updated the text to reflect this: “Because the T6IC1 putative immunity gene was predicted to encode a transmembrane anchoring domain with the bulk of the protein facing the periplasm, we hypothesized that the activity of T6EC1 is in the periplasm (Supplementary Figure 3A)”. Furthermore, in Supplementary Figure 3A, I added the details in the legend: “The transmembrane prediction is as follows: residues 1-6 are inside, residues 7-26 are transmembrane helices, and residues 27-157 are outside (i.e. in the periplasm).” I’m glad your close reading of the paper caught this.

(Interestingly, as a side note, the canonical colicin M does not seem to have activity when overproduced in the cytoplasm [PMC8469651], so the simplest expectation would be that it is

the same for T6EC1. While canonical colicin M is taken in by a membrane bound receptor, T6SS effectors have the ability to penetrate membranes, so perhaps T6EC1 has different constraints and potential cytosolic activity... an interesting topic for a future study)

The Integrated Microbial Genomes and Microbiomes should be named IMG/M as indicated on their website and not IMG.

Fixed.

Lines 341-342 are repetitive to lines 342-344.

The extra sentence was removed.

Line 403, the reference to Figure 4D is missing.

Fixed.

Reviewer #2:

The manuscript "Identification of type VI secretion system effector-immunity pairs using structural bioinformatics" from Geller et al. is interesting and pleasant to read. The work presents an interesting approach to predict protein-protein interactions in bacteria based on AlphaFold2 and FoldSeek, which are newborn technologies in the structural bioinformatics field. In particular, here the authors focus on the proteins belonging to the T6SS, an important bacterial machinery able to inject effectors into preys. The manuscript describes an in silico approach to detect bacterial proteins secreted by the T6SS and the associated immunity proteins that protect the bacteria from these proteins. The method allows functional annotation of effectors and detection of the immunity partner, and finally the database is shared with the community. Some selected predictions are experimentally validated. The authors focus on 'specialized' effectors, which are fused with components of the T6SS. Using sequence analysis, they extensively identify the specialized effectors in bacterial genomes. Structures of extra-domain were predicted with AlphaFold2 and the structural comparison with Foldseek was carried out to suggest functional annotations. Immunity proteins were predicted based on their genomic location (downstream and in the vicinity of effectors) and the AlphaFold prediction of the effector/immunity protein dimer.

Experimental validation was carried out on four pairs of effector + immunity proteins.

The main ideas and hypotheses are clearly stated. The results support the claim.

The findings will be of interest for the community of microbiologists. Here I list comments to improve the methods.

Thanks for reading the manuscript, and I am glad to hear you think it will be interesting for the broader community.

A possible validation of the classifier that they developed is to see how it behaves against noise. Thus I wonder what happens if you take all your known effector-immunity pairs and randomize them (ie, you randomize the pairing between the effector and immunity protein), and see what is the resulting ipTM score for the randomized pairs, and how they are classified. I guess ipTM should be low, but it is worth seeing to have additional trust on the classifier. Furthermore, it would be nice to see the same analysis for a set of predicted effector-immunity pairs, ie, randomize the pairs as described above.

I love the logic here and because of that, this is precisely what I thought would be the best training data for the classifier. To be sure I don't feed garbage data into my classifier, I trained it using known T6E-immunity pairs from the SecReT6 database, and I specifically chose those that have experimental validation. For a negative set, I indeed did as you suggest, I randomized the T6E-immunity pairs so they are not supposed to bind one another. Indeed, the ipTM scores were low, as you surmised, and led to good stats for the training of a model seen in Supplementary Figure 2A. In Figure 2B, you can see that the same analysis for the predicted pairs, i.e. those with no experimental validation, and that the classifier does an excellent job separating the randomized pairs from the cognate pairs. An interesting question about this classifier is whether perhaps it is applicable to all kinds of effector-immunity pairs from different secretion systems, and even to toxin-antitoxin pairs, which could be a nice topic for further study. In the future study, I would indeed use your same idea of randomization to test how well the ipTM score works to predict binding, too.

The structural comparison with FoldSeek is used to propose a functional annotation of the specialized effector extra domains. At the level of structural similarity, one can expect that a given structure can match with several proteins of different functions. It would be interesting to know how this multiplicity is handled.

I manually curated the top structural homolog "hits" from the structure-structure search process, and I saw that in many cases, it was clear that the protein had one very specific function. For example, the colicin M homolog discussed in the manuscript was a clear one-to-one hit with one possible annotation. However, as you suggest, there was also detection of homology between one query protein with multiple proteins with various annotations. This was usually because of homology of a domain in putative T6E with a small domain such as LysM, which is quite a widespread generalist domain and has unclear implications for T6E function. This LysM domain can therefore be a domain in many multi domain proteins, and these may have different annotations due to variance in the other domains. Since the E-values, RMSD, as well as the manual curations showed the homolog was trustworthy, I still wanted to list something as its annotation. I tried my best to be descriptive and as to not accidentally mislead members of the community, so for example, I would have just written the domain that it matches, LysM, rather than guessing which overall protein annotation is correct from the multiplicity of annotations. In the future, structural data will be more organized, clustered, and hierarchical, making it easier to give proper annotations in an organized fashion (indeed work like this has begun <https://www.nature.com/articles/s41586-023-06510-w>). Also automated annotation with the help of deep learning will be able to systematize tasks like this that I did here manually, and this will

also help with consistency. Future studies may use tools like ProtelInfer (<https://elifesciences.org/articles/80942>) or later versions of them. I am optimistic about the future of these kinds of structure-structure comparison that will only get better as we collect more data and use novel tools to organize and analyze them.

The authors performed a mutation on a candidate effector, and state that "we performed a point mutagenesis of the catalytic glutamate to glutamine, causing minimal changes in steric structures". Although it seems safe to postulate that the mutation will not perturb the structure, I would suggest the authors to be less affirmative.

I understand that this sentence was a bit overstated, as I was genuinely excited at how simply changing an oxygen to a nitrogen could phenotypically have such a dramatic effect! But I do agree we don't really know in any detail about steric effects, so I changed it to "To test this, we performed a point mutagenesis of the catalytic glutamate to glutamine (E108Q), which makes a minimal change to the side-chain structure, yet changes its chemical nature".

Minor points

line 79 I think it is generally accepted to say that AF2 didn't solve the protein folding problem, but improved protein structure prediction, only for folded domains. Protein folding is a more complex problem involving the dynamics and the pathway of the process, which is still not systematically resolved.

Good point. As you say, the protein folding problem has many parts: "The 'protein folding problem' consists of three closely related puzzles: (a) What is the folding code? (b) What is the folding mechanism? (c) Can we predict the native structure of a protein from its amino acid sequence?" (PMID 18573083). There are definitely a lot of nuances here that I did not get into since it is outside of the scope of the paper. I changed the text to reflect a breakthrough in "protein structure prediction" rather than to the protein folding problem as a whole.

line 97 thier -> their

Fixed.

The above evaluation is about the in silico approach, and does not concern the experimental validation.

Reviewer #3:

In this manuscript, the authors present a large scale integrated bioinformatic predictive structural analysis of effector and immunity genes of the type VI secretion system of the phylum

Pseudomonodota. This pipeline leverages validated effector and immunity gene pairs to discover new pairs. Further, the authors validate the toxicity of 4 new previously unexplored effectors and demonstrate the neutralization capacity of the associated immunity genes. All together, the paper is well written and interesting, the method is powerful and provides more comprehensive insight into the spectrum of T6SS genes. Generally, I like the paper and I only have a few comments, which I list below.

Thanks for your review, and it is good to hear you thought it was interesting and that the method was powerful!

My major comment is that I request that the authors significantly tone down the strength of language of their interpretations throughout the entire manuscript. The alphafold modeling, while powerful, is prediction only and should not be overinterpreted. For example, the RMSD values of the T6E1-4 effectors is somewhat poor. While these values reflect structural similarity across the whole proteins, the authors do not specifically compare in this structural analysis potential enzymatic active sites (e.g. colicin M <https://doi.org/10.1074/jbc.M109.093583>) and, lacking this comparison and followup experimental mutational analysis of potential active sites in T6E1-4, I think the authors need to make abundantly clear that effector mechanism is unknown and therefore speculation about mechanism should be clearly conveyed as it is - speculation.

I agree that this manuscript is not meant to make any definitive interpretations, especially regarding the mechanism of actions of the putative effectors. I tried to be careful to emphasize that this paper was mainly about the use of the tools and how they are a significant improvement upon sequence-based methods. For example, in the discussion of the submitted manuscript, I wrote, "We emphasize that the main focus of this study was to explore the use of novel tools to see how they could augment T6E discovery and functional characterization. Future work beyond the scope of this manuscript will explore each of these T6E structural clusters for a higher-resolution understanding of their activities and of their mechanisms of action."

I understand your concerns, and in order that there is no doubt to the reader, I added more statements to this section of the manuscript to make it crystal clear. The updated statement is: "We emphasize that the main focus of this study was to explore the use of novel tools to see how they could augment T6E discovery and functional characterization. We were careful to note that Foldseek-based annotation is useful for generating accurate *hypotheses* of mechanisms of action. We sought to perform simple experiments to show that empirical data supports, rather than falsifies, our hypotheses. The experiments presented in this study are indeed in line with the hypotheses generated with the help of Foldseek, but are not meant to be exhaustive. Future work beyond the scope of this manuscript will explore each of these T6E structural clusters for a higher-resolution understanding of their activities and of the specific molecular determinants of their mechanisms of action."

Beyond the discussion, I made many edits in the paper, for example I added: "Further experiments are needed to confirm the details of the mechanism of action and that the putative

effector is indeed secreted in a T6SS-dependent manner.” Another example of a similar statement that I added: “Although further *in vitro* experiments on purified protein must be performed to ultimately confirm the mechanism of action, this result is in line with the hypothesis that T6EC4 is an endonuclease” amongst other edits.

As you rightly mentioned, the next experimental step would be to look at conserved residues that perhaps represent potential active sites for mutational analysis, which you can see in Figure 2C. The yellow residues are conserved, and interestingly seem to all be occluded by the putative immunity protein (according to the alphafold-multimer model). In the case of T6EC3, which I thought was the most interesting of the effectors, I did take the next step of doing an experimental mutational analysis of a potential active site. I was happy to see that it seems that T6EC3’s homologous putative catalytic residue is necessary for toxic activity (Figure 4D). In a future experimental study, I agree the next steps would be to explore more mutations of putative catalytic active site residues, and to do *in vitro* assays if possible to understand the targeting. It would be good to do this for the other effectors, which I think can be the focus of future work(s).

The authors search for "core" proteins (Paar, vgrg, hcp, etc) but not for structural T6SS proteins. It seems possible that many genomes lack functional T6SS and I would think that assessing if the entire repertoire of T6SS structural proteins is found encoded in the genomes examined would be very important. Just as an example, in Lines 504-513, the authors discuss effectors that lack immunity, which is counter to expectations. In these genomes, are structural genes missing? Another possibility, unless I misunderstand the analysis described on lines 239-243 in which I think only looked for immunity downstream of the effector, is that the immunity gene is actually upstream of the effector

I was also concerned, as you are, that looking only at “core” proteins may not be specific enough for the analysis, so I only analyzed genomes encoding for T6SS overall. I defined all the genomes in the analysis as T6SS-encoding genomes if they have at least eight T6SS structural domains, which are listed in a Supplementary Table 2 (for example, pfam12790 representing TssJ, IPR010263 representing TssK, etc.). Therefore, I believe that the genomes can safely be defined as encoding for full T6SS operons, at least for this specific analysis.

In the discussion, I mention a few biological reasons we may not be finding immunity proteins for each effector, chiefly among them is that the effectors are anti-eukaryotic. Further reasons include generalized immunity mechanisms like cell wall modification, immunity islands, and stress responses (Hersch et al, 2020a and 2020b; Le et al, 2020; Ross et al, 2019). In terms of the parameters used in the analysis, I was extra conservative by following the model that immunity proteins are likely small proteins (<700 amino acids in length based on the SecReT6 database of immunity proteins), that are likely encoded downstream in the same “sense” on the DNA. Of course you are completely right that the immunity protein can be encoded upstream, but because the main goal of the analysis was to show that alphafold-multimer could be used to quantify the interaction between a putative effector and its cognate immunity protein, I did not want to risk adding any upstream gene that does not follow the canonical model of effector

immunity genomic organization. Now that we see the utility of alphafold-multimer, we and others in the community can indeed expand this analysis by looking upstream as well for immunity proteins if needed using the logistic regression model we developed.

The pie chart figure in 3B is unreadable. I recommend grouping/collapsing some of the categories into "other".

Okay, I did exactly as you suggested and added an "other" category for the unreadable parts.

There is no information on biological replicates performed for the toxicity experiments and no description of statistical tests used - the authors must add this information. Likewise, there is no quantitative information regarding data from microscopy experiments. There are many useful tools (e.g. microbeJ) that facilitate quantitative cell biological examination of microscopy datasets and I urge the authors to add this, otherwise the examples shown lack value.

I apologize for the oversight, I added the biological replicate information and a non-parametric statistical test as appropriate for the drop assay quantification in the figure legend (Figure 2D). I updated the materials and methods as well.

The microscopy does indeed have a quantification for rounding, in Supplementary Figure 4, using a program we developed based on the foundation model Cellpose (Stringer et al, 2021) that we called cellstats (<https://github.com/noamblum/cellstats>). We also performed the appropriate non-parametric test for differences in aspect ratio and found rounding was caused by the expression of the T6E. Unfortunately the DNA puncta was more difficult to quantify, but with further development of this tool, it may be possible. I can say that the microscopy images are representative of what we have seen overall, and some will be submitted to the journal for archiving of source data.

There is a tendency to cite recent reviews instead of primary literature and I encourage the authors to cite both. For example, the statement on line 62 regarding the role of immunity should include reference Hood et al PMID 20114026.

There is definitely a spectrum between using primary literature and using reviews, and I agree with your assessment that I tend to naturally use reviews. I went through and added more primary sources— including the one you suggest here— and I hope this re-balances my tendencies.

The authors should provide analysis code in a publicly accessible repository e.g. github.

Yes, agreed, so I added relevant scripts and commands here at https://github.com/alexlevylab/T6E_discovery and added this link to the materials and methods for anyone interested.

19th Mar 2024

Manuscript Number: MSB-2024-12200R

Title: Identification of type VI secretion system effector-immunity pairs using structural bioinformatics

Dear Dr Levy,

Thank you for the submission of your revised manuscript to Molecular Systems Biology. We have now received the enclosed reports from the referees that were asked to re-assess it. As you will see the reviewers are now globally supportive and I am pleased to inform you that we will be able to accept your manuscript pending the following final amendments:

1) In the main manuscript file, please do the following:

- Please include up to 5 keywords

- Please format the Data availability section according to the example below:

The computer code produced in this study is available in the following database:

- Modeling computer scripts: GitHub (<https://github.com/SysBioChalmers/GECKO/releases/tag/v1.0>)

- Please rename "Conflict of Interest Statement" to "Disclosure and competing interests statement". We updated our journal's competing interests policy in January 2022 and request authors to consider both actual and perceived competing interests. Please review the policy <https://www.embopress.org/competing-interests> and update your competing interests if necessary.

2) In the Materials and Methods, please take care of the following:

- Please ensure that a statement on whether or not blinding was done is included in the Materials and Methods even if no blinding was done. Please also be sure to update the Author Checklist, indicating that this has been included in the manuscript.

3) Please place individual sections of the manuscript in the following order: Title page - Abstract & Keywords - Introduction - Results - Discussion - Materials & Methods - Data Availability - Acknowledgements - Disclosure and Competing Interests Statement - References - Figure Legends - Tables - Expanded View Figure Legends.

- The main and EV figure legends need to be moved to come after the References.

4) For the figures and figure legends, please take care of the following:

- Please note that information related to n is missing in the legend of Figure EV 2b.

- Please note that we require exact p-values to be reported. Currently exact p-values are not provided in Figure EV 4a or its legend.

- Please rename the movie to Movie EV1 and update its callout in the main manuscript text. The legend should be removed from the main manuscript and should be included as a separate file zipped together with the movie.

5) Tables: Please rename Tables EV1-EV2 to Dataset EV1-EV2. Each dataset will need its legend removed from the manuscript and added to the corresponding file in a separate tab. Please update their callouts in main manuscript text.

6) Synopsis:

- Synopsis image: Please upload the image separately (not in the manuscript) as a high-resolution jpeg file 550 pixels wide x (250-400) pixels high. Currently the image file is too large.

- Synopsis text: Please provide a short standfirst (maximum of 300 characters, including space), limit the bullet points to max. 5 and upload it as a separate .doc file. Please write the bullet points to summarise the key NEW findings. They should be designed to be complementary to the abstract - i.e. not repeat the same text. We encourage inclusion of key acronyms and quantitative information (maximum of 30 words / bullet point). Please use the passive voice.

7) Source Data: Please ensure that your source data are uploaded as a single source data file (zipped) per figure, with the panels clearly visible in the folder structure. e.g. all the Source data files for figure 4 need to be saved in a single folder and this needs to be zipped and then uploaded as "SD figure 4.zip" file.

8) Please update the README file available on Github or provide a new file with practical use instructions for potential future users of your code.

9) As part of the EMBO Publications transparent editorial process initiative (see our policy here:

https://www.embopress.org/transparent-process#Review_Process), Molecular Systems Biology will publish online a Peer Review File (PRF) to accompany accepted manuscripts. This file will be published in conjunction with your paper and will include the anonymous referee reports, your point-by-point response and all pertinent correspondence relating to the manuscript. Let us know whether you agree with the publication of the PRF and as here, if you want to remove or not any figures from it prior to publication. Please note that the Authors checklist will be published at the end of the PRF.

10) Please provide a point-by-point letter INCLUDING my comments as well as the reviewer's reports and your detailed responses (as Word file).

I look forward to reading a new revised version of your manuscript as soon as possible.

Yours sincerely,

Poonam Bheda, PhD

Scientific Editor
Molecular Systems Biology

Please click on the link below to submit the revision online:

Reviewer #1:

The authors have done an excellent job reviewing the manuscript. I am happy with their responses to the concerns and suggestions raised during the review process. I have no further comments to add but to congratulate the authors for a great work.

A minor thing: Line 540 - please use "Hcp" instead of "HCP"

Reviewer #2:

The reviewers responded to all my questions.

Reviewer #3:

I thank the authors for responding to my initial review in detail and I am largely satisfied with their edits and additions. I recommend using terminology such as "neutralize" or "prevent toxicity" instead of "save", which occurs periodically throughout the manuscript.

19th Mar 2024

Manuscript Number: MSB-2024-12200R

Title: Identification of type VI secretion system effector-immunity pairs using structural bioinformatics

Dear Dr Levy,

Thank you for the submission of your revised manuscript to Molecular Systems Biology. We have now received the enclosed reports from the referees that were asked to re-assess it. As you will see the reviewers are now globally supportive and I am pleased to inform you that we will be able to accept your manuscript pending the following final amendments:

We are glad to hear this good news!

1) In the main manuscript file, please do the following:

- Please include up to 5 keywords

I added 5 keywords below the abstract, separated by slashes, as per the instructions for authors.

- Please format the Data availability section according to the example below:

The computer code produced in this study is available in the following database:

- Modeling computer scripts: GitHub

<https://github.com/SysBioChalmers/GECKO/releases/tag/v1.0>

I re-formatted this section using the phrasing that you provided here.

- Please rename "Conflict of Interest Statement" to "Disclosure and competing interests statement". We updated our journal's competing interests policy in January 2022 and request authors to consider both actual and perceived competing interests. Please review the policy <https://www.embopress.org/competing-interests> and update your competing interests if necessary.

I updated the statement as requested.

2) In the Materials and Methods, please take care of the following:

- Please ensure that a statement on whether or not blinding was done is included in the Materials and Methods even if no blinding was done. Please also be sure to update the Author Checklist, indicating that this has been included in the manuscript.

I included a statement on blinding ("Blinding was not employed as it was deemed not relevant to the experimental design and objective") and I updated the Author Checklist accordingly.

3) Please place individual sections of the manuscript in the following order: Title page - Abstract & Keywords - Introduction - Results - Discussion - Materials & Methods - Data Availability - Acknowledgements - Disclosure and Competing Interests Statement - References - Figure Legends - Tables - Expanded View Figure Legends.

- The main and EV figure legends need to be moved to come after the References.
I placed the sections in the order as instructed, including moving the Figure and EV Figure legends to after the references.

4) For the figures and figure legends, please take care of the following:

- Please note that information related to n is missing in the legend of Figure EV 2b.

I added the relevant information including k and n: "Bars represent mean, error bars represent standard deviation of k = 10 instances of k-fold validation performed on a dataset of n=164 pairs (95 in positive set, 69 in the negative set)"

- Please note that we require exact p-values to be reported. Currently exact p-values are not provided in Figure EV 4a or its legend.

In the previous draft, I reported the p-value previously as equal to zero, as the number was so small, the computer's floating point minimum was reached. In order to comply with your request, I used logarithms to calculate the order of magnitude of the p-value, which is 1E-1871. I wrote this p-value into Figure EV 4a.

- Please rename the movie to Movie EV1 and update its callout in the main manuscript text. The legend should be removed from the main manuscript and should be included as a separate file zipped together with the movie.

Okay, I changed the name and removed the legend from the main manuscript.

5) Tables: Please rename Tables EV1-EV2 to Dataset EV1-EV2. Each dataset will need its legend removed from the manuscript and added to the corresponding file in a separate tab. Please update their callouts in main manuscript text.

Done.

6) Synopsis:

- Synopsis image: Please upload the image separately (not in the manuscript) as a high-resolution jpeg file 550 pixels wide x (250-400) pixels high. Currently the image file is too large.

Okay I uploaded it separately in the size and format requested.

- Synopsis text: Please provide a short standfirst (maximum of 300 characters, including space), limit the bullet points to max. 5 and upload it as a separate .doc file. Please write the bullet points to summarise the key NEW findings. They should be designed to be complementary to the abstract - i.e. not repeat the same text. We encourage inclusion of key acronyms and quantitative information (maximum of 30 words / bullet point). Please use the passive voice.

I wrote the standfirst and have it in a separate file.

7) Source Data: Please ensure that your source data are uploaded as a single source data file (zipped) per figure, with the panels clearly visible in the folder structure. e.g. all the Source data

files for figure 4 need to be saved in a single folder and this needs to be zipped and then uploaded as "SD figure 4.zip" file.

Okay, I put related figure source data together into zipped format.

8) Please update the README file available on Github or provide a new file with practical use instructions for potential future users of your code.

I added a section to the README section, describing the inputs and outputs of the relevant scripts and their dependencies. This should make the pipeline accessible to future users.

9) As part of the EMBO Publications transparent editorial process initiative (see our policy here: https://www.embopress.org/transparent-process#Review_Process), Molecular Systems Biology will publish online a Peer Review File (PRF) to accompany accepted manuscripts. This file will be published in conjunction with your paper and will include the anonymous referee reports, your point-by-point response and all pertinent correspondence relating to the manuscript. Let us know whether you agree with the publication of the PRF and as here, if you want to remove or not any figures from it prior to publication. Please note that the Authors checklist will be published at the end of the PRF.

Yes, you can publish the PRF

10) Please provide a point-by-point letter INCLUDING my comments as well as the reviewer's reports and your detailed responses (as Word file).

Okay I integrated your comments, as well as our cover letter in response to your comments, and the point by point response.

I look forward to reading a new revised version of your manuscript as soon as possible.

Yours sincerely,

Poonam Bheda, PhD
Scientific Editor
Molecular Systems Biology

Reviewer #1:

The authors have done an excellent job reviewing the manuscript. I am happy with their responses to the concerns and suggestions raised during the review process. I have no further comments to add but to congratulate the authors for a great work.

I am glad we satisfied your concerns and suggestions, and I appreciate the congratulations!
Thanks for the detailed review!

A minor thing: Line 540 - please use "Hcp" instead of "HCP"
Good catch – Changed.

Reviewer #2:

The reviewers responded to all my questions.
Great, thank you for your helpful review!

Reviewer #3:

I thank the authors for responding to my initial review in detail and I am largely satisfied with their edits and additions. I recommend using terminology such as "neutralize" or "prevent toxicity" instead of "save", which occurs periodically throughout the manuscript.

Glad to hear you're largely satisfied with the edits and additions, thanks for your careful review!

I changed the term "saved" in figure 2 legend, as well as in the main text.

9th Apr 2024

Manuscript number: MSB-2024-12200RR

Title: Identification of type VI secretion system effector-immunity pairs using structural bioinformatics

Dear Dr Levy,

Thank you again for sending us your revised manuscript. We are now satisfied with the modifications made and I am pleased to inform you that your paper has been accepted for publication.

Yours sincerely,

Poonam Bheda, PhD
Scientific Editor
Molecular Systems Biology
